# Loss of wild-type p53 promotes mutant p53-driven metastasis through acquisition of survival and tumor-initiating properties

Mizuho Nakayama[1,2], Chang Pyo Hong[3], Hiroko Oshima[1,2], Eri Sakai[1], Seong-Jin Kim[3,4] & Masanobu Oshima [1,2✉]

Missense-type mutant p53 plays a tumor-promoting role through gain-of-function (GOF) mechanism. In addition, the loss of wild-type *TP53* through loss of heterozygosity (LOH) is widely found in cancer cells. However, malignant progression induced by cooperation of *TP53* GOF mutation and LOH remains poorly understood. Here, we show that mouse intestinal tumors carrying *Trp53* GOF mutation with LOH (AKTP^M/LOH) are enriched in metastatic lesions when heterozygous *Trp53* mutant cells (AKTP^+/M) are transplanted. We show that *Trp53* LOH is required for dormant cell survival and clonal expansion of cancer cells. Moreover, AKTP^M/LOH cells show an increased in vivo tumor-initiating ability compared with AKTP^Null and AKTP^+/M cells. RNAseq analyses reveal that inflammatory and growth factor/ MAPK pathways are specifically activated in AKTP^M/LOH cells, while the stem cell signature is upregulated in both AKTP^M/LOH and AKTP^Null cells. These results indicate that *TP53/Trp53* LOH promotes *TP53/Trp53* GOF mutation-driven metastasis through the activation of distinct pathway combination.

[1] Division of Genetics, Cancer Research Institute, Kanazawa University, Kanazawa 920-1192, Japan. [2] WPI Nano Life Science Institute, Kanazawa University, Kanazawa 920-1192, Japan. [3] Theragen Etex Bio Institute, Suwon 16229, Republic of Korea. [4] Precision Medicine Research Center, Advanced Institute of Convergence Technology and Department of Transdisciplinary Studies, Seoul National University, Suwon 16229, Republic of Korea. ✉email: oshimam@staff.kanazawa-u.ac.jp

Colorectal cancer (CRC) is a leading cause of cancer-related death around the world[1,2]. The most frequently mutated gene in the pan-cancer cohort is *TP53*[3], and its mutations in CRC are detected in about 55–60% of cases[4,5]. During the malignant progression of CRC, *TP53* mutations occur near the transition from benign to malignant lesion[6], and indeed, the mutation incidence was shown to be about 80% when metastasis-associated CRCs were examined[7]. These results suggest that *TP53* mutations play a role in the promotion of malignant progression in CRC.

Unlike other tumor suppressor genes, the majority of *TP53* mutations are missense-type at hot spots, resulting in the expression of mutant p53 protein with a single amino acid substitution[8,9]. It has been shown that such mutant p53 plays an oncogenic role through a gain of function (GOF) mechanism. For example, mouse models expressing mutant p53R172H and p53R270H (mutation at codons 175 and 273 in humans) developed adenocarcinomas in the intestine and lung that were not found in *Trp53*-disrupted p53Null mice[10,11]. In addition, p53R172H mice demonstrated metastasis of pancreatic cancer[12], and we recently showed that the expression of p53R270H induces the submucosal invasion of intestinal tumors in an *Apc*Δ716 mouse model[13,14]. Importantly, the ablation of mutant p53 expression in cancer cells suppressed transplanted tumor growth in vivo and extended the animal survival, indicating that tumor growth is dependent on the sustained expression of mutant p53[15]. Mechanically, it has been shown that the expression of mutant p53 results in expansion of mammary epithelial stem cells[16] and that mutant p53 induces stem cell gene signatures in CRC as well as mesenchymal stem cell-derived tumors[17,18]. These results suggest that mutant p53 promotes the late stage of tumorigenesis, possibly through the acquisition of an invasive ability and stem cell characteristics.

Several molecular mechanisms underlying the involvement of mutant p53 in malignant progression have been reported, including constitutive activation of integrin and epidermal growth factor receptor (EGFR) signaling and the activation of TGF-β-dependent migration and PDGF receptor signaling[19–21]. In addition, it was recently shown that mutant p53 induces global transcriptional shift by epigenetic switching through interaction with the chromatin remodeling complex or the modification of histone methylation and acetylation[22,23]. In addition to these acquired oncogenic functions of mutant p53, the loss of wild-type p53 through the loss of heterozygosity (LOH) is found in >93% of human cancers[24]. This loss also plays an important role in malignant progression. We and other groups have shown that *Trp53* LOH is important for the stabilization and nuclear accumulation of the mutant p53[13,14,25]. However, the in vivo mechanism underlying the combination of the expression of GOF mutant p53 and loss of wild-type p53 by LOH for malignant progression is poorly understood.

We previously generated an intestinal tumor metastasis model by splenic transplantation of mouse intestinal tumor-derived organoids, termed AKTP+/M cells, that carry *Apc*Δ716, *Kras*+/G12D, *Tgfbr2*-/- and *Trp53*+/R270H mutations simultaneously[26]. These four-driver genes are included among the frequently mutated genes in human CRC[3,4] and are well-characterized as genes responsible for the promotion of CRC multistep tumorigenesis[27]. In the present study, we investigate the role of the loss of wild-type *Trp53* by LOH in the liver metastasis of AKTP+/M cells carrying a heterozygous *Trp53* GOF mutation. We report that *Trp53* LOH in combination with the expression of GOF mutant p53 is required for the survival of disseminated cancer cells and subsequent clonal expansion, which leads to metastasis development. We also show that inflammatory and MAPK pathways in addition to the stem cell pathway are activated in AKTPM/LOH cells. These results provide a mechanism involving GOF mutant p53 together with loss of wild-type p53 for acceleration of metastasis, findings that will contribute to the future development of therapeutic strategies against CRC metastasis.

## Results

**Enrichment of *Trp53* LOH cells in liver metastasis tumors**. We previously generated intestinal tumor-derived organoid cells (AKTP+/M cells) from compound mutant mice, in which genetic alterations were simultaneously introduced to four colon cancer driver genes: *Apc*, *Kras*, *Tgfbr2*, and *Trp53*[26]. The genotype of *Trp53* in AKTP+/M cells is heterozygous (+/R270H), which is one of *Trp53* GOF mutations[8,9]. When AKTP+/M cells were transplanted to the spleen of immunodeficient mice (Fig. 1a), multiple metastatic tumors developed in the liver with 100% incidence, which is consistent with the previous report[26] (Fig. 1b). Histologically, the metastasized tumors in the liver showed an advanced malignant phenotype compared with the primary tumors in the spleen, including an epithelial-mesenchymal transition (EMT)-like structure and the generation of fibrotic microenvironment with α-smooth muscle actin (SMA)-positive myofibroblasts and the deposition of collagen fibers in the stroma (Fig. 1c). These results suggest that transplanted tumor cells acquired malignant characteristics during the metastasis process.

In the human CRC cells carrying *TP53* mutations, loss of wild-type *TP53* by LOH is frequently found[4,24], suggesting that the combination of fa *TP53* GOF mutation and the subsequent loss of wild-type *TP53* is important for the acquisition of a malignant phenotype. To determine whether or not the wild-type *Trp53* gene was lost in the metastasized AKTP cells, we collected tumor cells from paraffin-embedded sections of the spleen and liver tissues through laser microdissection (Supplementary Fig. 1) and performed *Trp53* allele-specific genomic polymerase chain reaction (PCR). We confirmed that the PCR system detected both mutant and wild-type *Trp53* simultaneously (Fig. 1d left). Of note, the relative band intensities of the wild-type *Trp53* were decreased significantly in 4 out of 8 liver metastasized tumor samples compared with that in the parental organoid cells, while those in all spleen tumor samples were at a similar level to the control organoid, indicating that the tumor cells that carried GOF mutant *Trp53* and lost wild-type *Trp53* (hereafter AKTPM/LOH cells) were enriched in the metastatic tumors. (Fig. 1d right, e).

It has been shown that loss of wild-type *Trp53* enhances the stabilization and nuclear accumulation of mutant p53[13,25]. Notably, the ratios of p53 nuclear accumulation without cytoplasmic distribution (nuclear-only accumulation) were higher in the liver metastasized tumor cells than in the primary spleen tumor cells (Fig. 1c, f). We further confirmed that the number of tumor glands with increased p53-nuc only cells (>80%) was significantly higher in the liver tumors than in the spleen tumors (Fig. 1g).

**Trp53 LOH for clonal expansion of dissociated single cells**. During the metastasis process, disseminated tumor cells are exposed to cellular stress, such as a dormant state and clonal proliferation[28]. To examine whether such stresses constitute selection pressure for the enrichment of AKTPM/LOH cells in the metastatic foci, we performed in vitro subcloning experiments (Fig. 2a). We confirmed that wild-type *Trp53* remained in AKTP+/M cells when cells were cultured in Matrigel with serial passages by mechanical dissociation with pipetting, with cells remaining as tumor glands rather than dissociating into single cells (Fig. 2a, b). In contrast, when AKTP+/M cells were enzymatically dissociated by trypsin to single cells and subjected to subcloning on the culture dish, all subclones derived from the

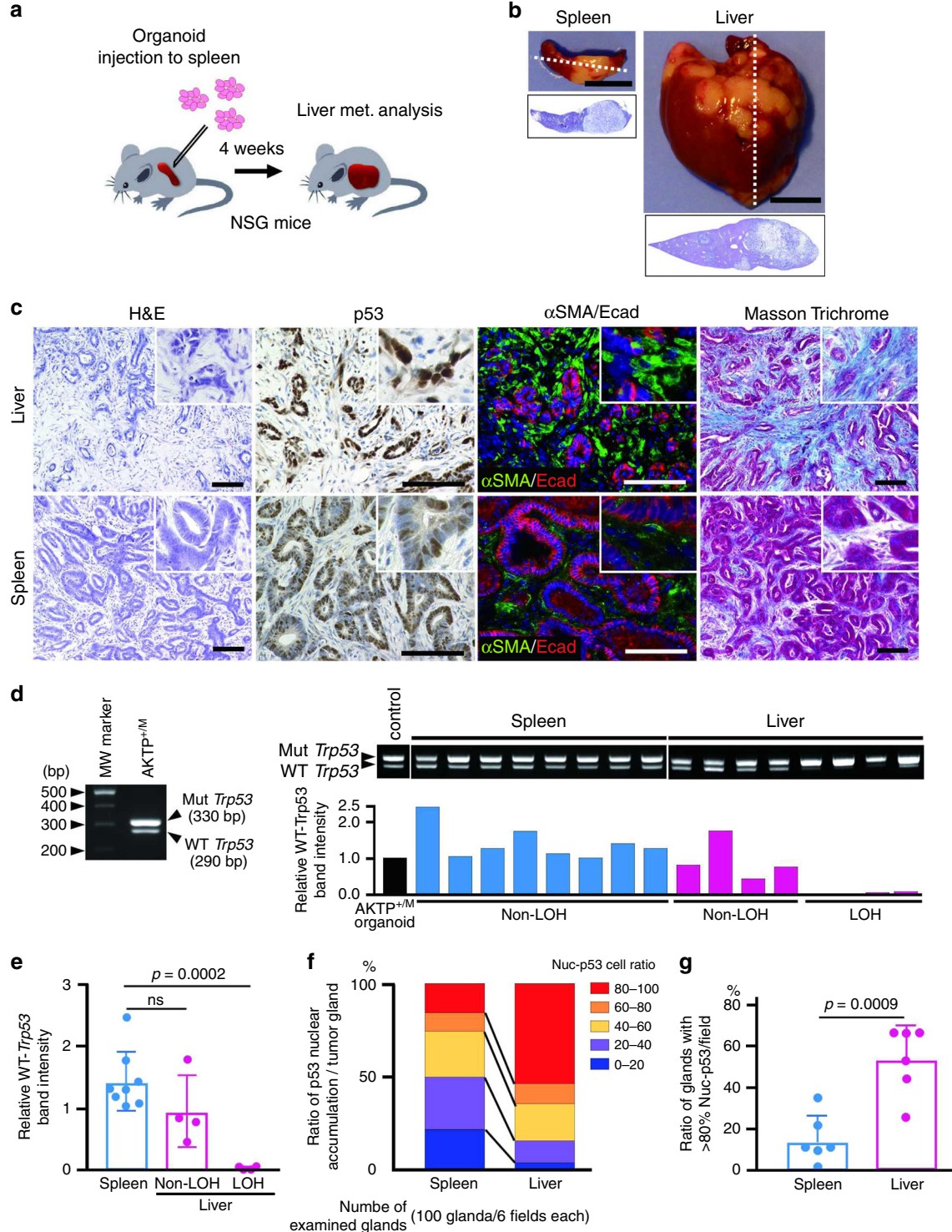

surviving cells showed loss of wild-type *Trp53*, indicating that *Trp53* LOH is required for the survival and clonal expansion of single dissociated cells (Fig. 2c). In addition, single-cell sub-cloning of ATP[+/M] and AKP[+/M] cells that carried the heterozygous *Trp53* GOF mutation in addition to *Apc Tgfbr2* and *Apc Kras* compound mutations, respectively, also lost wild-type *Trp53* in all surviving and expanded cell clones at 100% frequency (Fig. 2c). We confirmed that the rate of p53 nuclear-only accumulation was significantly higher in AKTP[M/LOH] organoid cells than in AKTP[+/M] cells (48.0% and 6.8%, respectively). In contrast, the majority of AKTP[+/M] cells showed nuclear/

cytoplasmic (73.0%) or cytoplasmic-only p53 distribution (11.6%), which were rarely found in AKTP[M/LOH] cells (Fig. 2d, e).

We next examined whether or not *Trp53* LOH is required for the survival and clonal expansion in three-dimensional (3D) culture conditions. Consistent with two-dimensional (2D) culture experiments, trypsin-dissociated single AKTP[M/LOH] cells but not AKTP[+/M] cells formed organoids in collagen gel (Fig. 2f left). Although a small number of AKTP[+/M] cells survived and formed organoids in Matrigel, the organoid formation efficiency was significantly higher in AKTP[M/LOH] cells (Fig. 2f right). It is possible that signaling from the extracellular matrix (ECM) in

**Fig. 1 Enrichment of _Trp53_ LOH cells in liver metastasized tumor cells. a** Strategy for liver metastasis development in NSG mice. **b** Representative photographs of the spleen and liver at 4 weeks after transplantation, and histology sections of dashed lines (H&E). Bars, 10 mm. **c** Representative histology of liver metastases (top) and primary spleen tumors (bottom). H&E, immunohistochemistry for p53, double-fluorescence immunostaining for α-smooth muscle actin (αSMA) and E-cadherin (Ecad) and Masson Trichrome staining (left to right). The insets show enlarged images. Bars, 100 μm. **d** Representative results of genomic polymerase chain reaction (PCR) for _Trp53_ in AKTP$^{+/M}$ cells (left) and spleen and liver tumors (right top). Each lane indicates the tumor cells collected from independent tumors in the spleen and liver. Mut _Trp53_ and WT _Trp53_ indicate mutant _Trp53_ R270H and wild-type _Trp53_, respectively. The AKTP$^{+/M}$ organoid DNA was used as control. The band intensities (WT _Trp53_/Mut _Trp53_) were quantified using ImageJ, and relative band intensities of each sample in spleen (blue bars) and liver (magenta bars) to that of control (closed bar) are indicated as bar graph ($n = 1$ for each bar corresponding to each band) (bottom). LOH, loss of heterozygosity. **e** The mean relative band intensities in spleen tumors (blue) and liver tumors (magenta) indicated in (**d**) are shown. Note that the level of the mean band intensity was significantly decreased in LOH tumors in the liver. **f** The ratio of tumor cells with p53 nuclear-only accumulation in the spleen and liver tumors are indicated by different colors, blue, 0–20%; purple, 20–40%: yellow, 40–60%, orange, 60–80%, and red, 80–100%. The nuclear accumulation of p53 was examined in 100 tumor glands (>20 cells/gland) for 6 microscopic fields. **g** The mean ratios of tumor glands containing nuclear-only p53 cells at >80% in spleen tumors (blue) and liver tumors (magenta) are shown. The data in **e**, **g** are presented as mean ± s.d. Two-sided unpaired _t_-test was used for statistical analysis. _p_ values are provided; ns, not significant. In **e**, $n = 8$ (spleen tumors) and $n = 4$ (non-LOH or LOH liver tumors) biologically independent samples. In **g**, $n = 6$ biologically independent samples. Source data are provided as a Source data file.

Matrigel enhances the dissociated cell survival. These results, taken together, indicate that _Trp53_ LOH is important for the dormant cell survival and clonal proliferation of cancer cells, possibly through the increased nuclear accumulation of mutant p53. In contrast, neither AKTP$^{M/LOH}$ and AKTP$^{+/M}$ cells developed colonies in soft agar (Fig. 2g), indicating that AKTP$^{M/LOH}$ cells have not yet acquired the ability to achieve anchorage-independent growth and thus may require signaling from the extracellular matrix (ECM) for clonal expansion.

Interestingly, the efficiency of the _Trp53_ LOH cell enrichment through the subcloning decreased to 40% when the _Fbxw7_ gene was disrupted in AKTP$^{+/M}$ cells (namely AKTFP$^{+/M}$ cells) (Fig. 2c). Of note, the genetic alterations in _TP53_ and _FBXW7_ in human CRC tend to be mutually exclusive, although the difference was not found to be significant (Supplementary Fig. 2). Accordingly, _Trp53_ LOH may not be essential for metastasis when the FBXW7 function is suppressed, although further investigations are needed.

**Mechanisms underlying _Trp53_ LOH in AKTP$^{+/M}$ cells.** We next confirmed the loss of wild-type _Trp53_ in AKTP$^{M/LOH}$ cells by direct sequencing, while both wild-type and mutant _Trp53_ were found in AKTP$^{+/M}$ cells (Fig. 3a). CMT93 mouse colon cancer cells were used for the _Trp53_ wild-type cell control. To further examine the LOH mechanism in AKTP$^{+/M}$ cells, comparative genomic hybridization (CGH) analyses were performed for three independent AKTP$^{M/LOH}$ lines: 1C9, 2A6, and 3F9 (Fig. 3b). We confirmed the euploidy status of parental AKTP$^{+/M}$ cells that were maintained in Matrigel by mechanical passages.

In contrast, common copy number variations were found in AKTP$^{M/LOH}$ lines, such as Chr4 loss in 1C9 and 3F9 cells, suggesting the advantages of these copy number variations for overcoming subcloning stress. Notably, a decrease in the copy number of Chr11 containing the _Trp53_ gene was detected in 3F9, indicating the hemizygous status of mutant _Trp53_ in this subclone (one _Trp53$^{R270H}$_ copy left in cells). Consistently, real-time genomic PCR using wild-type and mutant _Trp53_-specific primers revealed that 1C9 and 2A6 cells contained two copies of the mutant _Trp53_ gene, namely copy-number-neutral LOH (AKTP$^{M/M}$ cells), while 3F9 cells carried a single copy of the GOF mutant _Trp53_ (Fig. 3c). Notably, the protein levels of mutant p53 were similar in all three subclones (Supplementary Fig. 3). These results suggest that the copy number of the mutant p53 is not relevant for the survival and clonal expansion ability of cancer cells. For further analyses in this study, we used AKTP$^{M/M}$ cells as AKTP$^{M/LOH}$ cells.

**Protection from anoikis by _Trp53_ GOF mutation with LOH.** It has been shown that the disruption of cell-cell or cell-ECM contact causes intestinal epithelial cell death, known as anoikis[29]. p53 plays a role in the stress response through the induction of cell cycle arrest and apoptosis[30], so the loss of wild-type p53 helps protect the dissociated single cells from anoikis at least in part. We, therefore, examined the cellular response to anoikis of AKTP$^{M/LOH}$ cells. For this experiment, we additionally constructed AKTP$^{Null}$ cells, in which both alleles of _Trp53_ in AKTP$^{+/+}$ cells had been disrupted by the CRISPR/Cas9 system (Supplementary Fig. 4). AKTP$^{Null}$ cells developed cystic organoids in Matrigel, which were similar to AKTP$^{+/+}$ and AKTP$^{+/M}$ organoids (Fig. 4a). However, AKTP$^{M/LOH}$ formed complexed glandular structures, indicating that AKTP$^{M/LOH}$ cells acquired distinct characteristics from AKTP$^{+/M}$ and AKTP$^{Null}$ cells.

Flow cytometry analyses showed that the rate of apoptosis of AKTP$^{+/+}$ and AKTP$^{+/M}$ cells after trypsin-dissociation was similar (21.5% and 22.4%, respectively), indicating that the heterozygous _Trp53_ GOF mutation is not sufficient for protection from anoikis (Fig. 4b, Supplementary Fig. 5). As expected, the apoptosis rate of the trypsin-dissociated AKTP$^{Null}$ cells decreased significantly to 8.7%. Importantly, however, the apoptosis ratio was further suppressed to 1.0% in AKTP$^{M/LOH}$ cells, indicating that the combination of the _Trp53_ GOF mutation with LOH protects tumor cells from anoikis more efficiently than the biallelic loss of wild-type _Trp53_.

Consistently, AKTP$^{M/LOH}$ cells showed greater cloning efficiency than AKTP$^{Null}$ cells when a trypsin-dissociated single cell was seeded into each well of a 96-well plate (Fig. 4c). AKTP$^{+/+}$ and AKTP$^{+/M}$ cells failed to form colonies after single-cell dissociation in this assay. Similarly, the number of proliferated cells was significantly higher in AKTP$^{M/LOH}$ cells followed by AKTP$^{Null}$ cells compared with that in AKTP$^{+/+}$ and AKTP$^{+/M}$ cells when $5 \times 10^2$ single-dissociated cells were seeded into each well of a 96-well plate (Fig. 4d).

In contrast, when the mechanically dissociated cells were cultured in Matrigel after passage, the survival and growth rates of tumor cells were relatively unchanged among the all _Trp53_ genotypes, regardless of the _Trp53_ genotypes (Fig. 4e). These results, taken together, indicate that _Trp53_ LOH is not required for the survival and proliferation of tumor cells if cell-cell contacts are maintained as tumor glands.

**Tumor-initiation by _Trp53_ GOF mutation with LOH.** The acquisition of dormant cell survival and cloning expansion properties may be associated with the tumor-initiation capability.

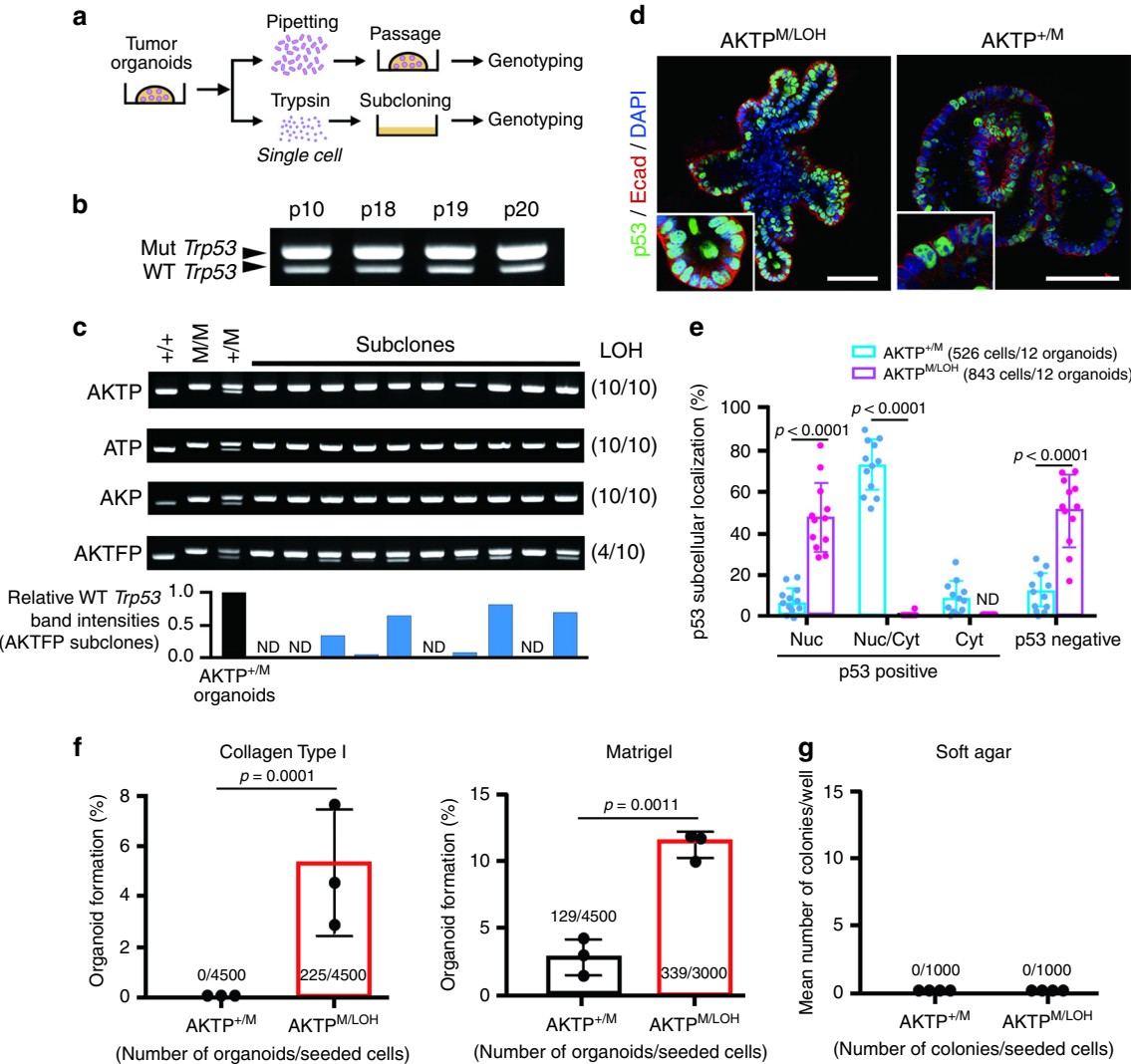

**Fig. 2 Selection of *Trp53* LOH cells in the subcloned intestinal tumor cells. a** Schematic drawing of the *Trp53* LOH analysis. The organoids were mechanically dissociated by pipetting and passaged into Matrigel or were enzymatically dissociated to single cells and subcloned on culture dishes. **b** Representative genomic PCR results of *Trp53* of AKTP$^{+/M}$ organoids maintained in Matrigel with mechanical passage (p, passage number). Mut *Trp53* and WT *Trp53* indicate mutant *Trp53* R270H and wild-type *Trp53*, respectively. Molecular weight marker for *Trp53* allele-specific PCR is shown in Fig. 1d. The experiments shown in (**b**) were repeated three times with similar results, and the representative results are shown. **c** Genomic PCR results of *Trp53* of independent subclones from AKTP$^{+/M}$, ATP$^{+/M}$, AKP$^{+/M}$, and AKTFP$^{+/M}$ organoids. Genomic DNAs of AKTP$^{+/+}$, AKTP$^{M/M}$, and AKTP$^{+/M}$ cells were used as controls. The numbers of *Trp53* LOH subclones per total subclones are indicated in parentheses. The band intensities (WT *Trp53*/Mut *Trp53*) of AKTPF subclones were quantified using ImageJ, and relative band intensities of each subclone (blue bars) to that of control (closed bar) are indicated as bar graph (*n* = 1 for each bar corresponding to each band) (bottom). ND, not detected. **d** Representative confocal images of AKTP$^{M/LOH}$ (left) and AKTP$^{+/M}$ (right) organoids immunostained for p53 and E-cadherin (Ecad) with DAPI nuclear staining are shown. The insets indicate enlarged images. Bars, 100 μm. **e** The percentages of p53 subcellular localization with nuclear-only (Nuc), mixed nuclear and cytoplasmic (Nuc/Cyt), and cytoplasmic-only (Cyt) in the AKTP$^{+/M}$ (blue) and AKTP$^{M/LOH}$ (magenta) organoid cells are shown. The total numbers of examined cells/organoids are indicated. **f** The ratios of organoid development from single AKTP$^{+/M}$ cells (closed bar) and AKTP$^{M/LOH}$ cells (red bars) in type I collagen gel (left) and Matrigel (right) are shown. The numbers of developed organoids and seeded cells are provided. **g** The results of soft agar colony formation assay. The number of colonies and seeded cells are provided. The data in **e**, **f** are presented as mean ± s.d., *n* = 12 (**e**) and *n* = 3 (**f**) biologically independent samples. *p* values are provided; ND, not detected. Two-sided unpaired *t*-test was used to calculate statistical difference. Source data are provided as a Source data file.

We, therefore, examined the in vivo tumor formation from a small number of AKTP cells with each *Trp53* genotype by subcutaneous (s.c.) transplantation of $1 \times 10^2$ trypsin-dissociated cells per site in NSG mice (Fig. 5a). AKTP$^{+/M}$ and AKTP$^{Null}$ cells developed 5 and 7 tumors, respectively, when injected at 14 sites (35.7 and 50%, respectively), while AKTP$^{+/+}$ cells did not form any tumors (Fig. 5b, c). Thus, either a heterozygous *Trp53* mutation or the loss of the wild-type p53 function contributes to an increased tumor-initiation ability. Notably, AKTP$^{M/LOH}$ cells induced tumor development in 12 out of 14 injected sites (85.7%

incidence), indicating that the combination of a *Trp53* GOF mutation with LOH further accelerates tumor initiation from single cells. However, when the respective *Trp53* genotype AKTP cells were mechanically dissociated (not to single cells) and transplanted at $5 \times 10^3$ cells per site, AKTP$^{+/M}$, AKTP$^{Null}$, and AKTP$^{M/LOH}$ cells induced tumor development at 11–12 of 15 injection sites (73.3–80%) (Fig. 5d, e). These results suggest that the combination of a *Trp53* GOF mutation with LOH is important for the development of tumor foci when a small number of single cells are disseminated.

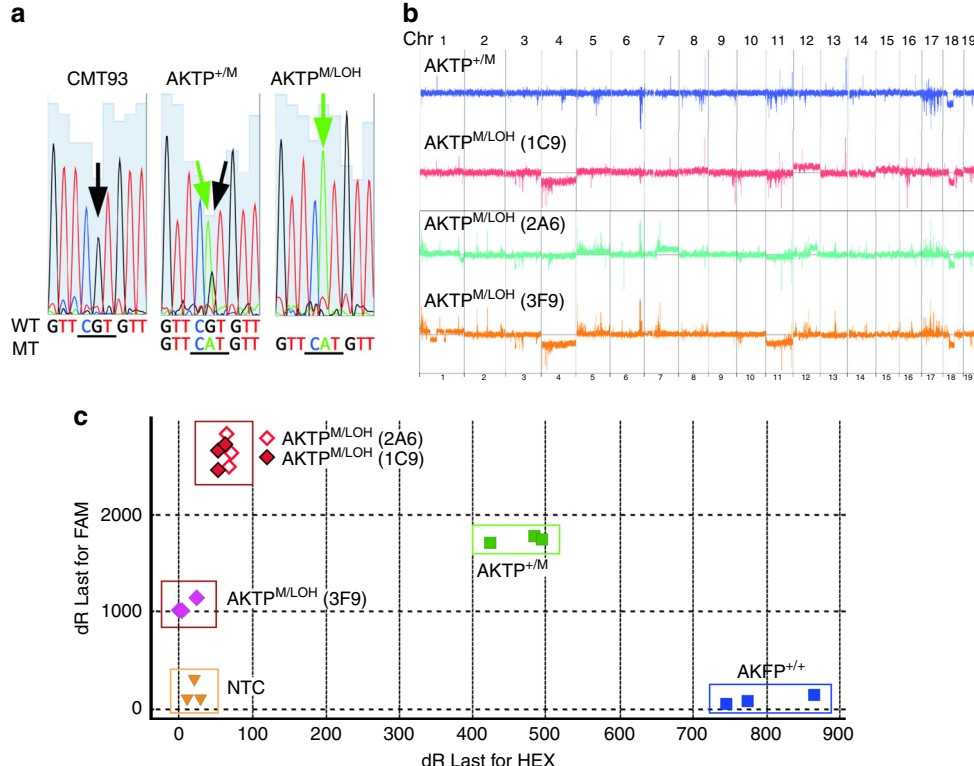

**Fig. 3 Mechanism underlying *Trp53* LOH in AKTP$^{+/M}$ cells. a** Direct sequencing of genomic DNA fragments including codon 270 (under line) of *Trp53* in CMT93 cells, AKTP$^{+/M}$, and AKTP$^{M/LOH}$ cells. WT, wild-type *Trp53*; and MT, mutant *Trp53* (R270H). **b** A whole-genome CGH analysis of the indicated AKTP$^{M/LOH}$ subclones (1C9, 2A6, and 3F9) and parental AKTP$^{+/M}$ cells. Chr indicates chromosome number. **c** TaqMan SNP genotyping results of AKTP$^{M/LOH}$ 2A6 cells (open diamonds with red line), AKTP$^{M/LOH}$ 1C9 cells (red diamonds with black line), AKTP$^{M/LOH}$ 3F9 cells (magenta diamonds), AKTP$^{+/M}$ cells (green squares), AKTP$^{+/+}$ cells (blue squares), and no template control (NTC) (yellow triangles) are shown as a scatter plot. The *Trp53* allelic discrimination data are shown as a scatter plot of *Trp53* wild-type codon 270 (CGT) (HEX dye) versus *Trp53* mutant codon R270H (CAT) (FAM dye). Note that two AKTP$^{M/LOH}$ subclones (2A6 and 1C9) carry two copies of mutant *Trp53* R270H, while one AKTP$^{M/LOH}$ subclone 3F9 carries one copy or *Trp53* R270H. Source data are provided as a Source data file.

Consistent with the liver metastasized tumors (Fig. 1c), AKTP$^{M/LOH}$ s.c. tumors showed a poorly differentiated EMT-like histology with the generation of a fibrotic microenvironment characterized by increased αSMA-positive myofibroblasts, which was more advanced than in AKTP$^{+/M}$ and AKTP$^{Null}$ cell tumors (Fig. 5f). However, no significant differences were noted in the Ki67-labeling indices among AKTP$^{+/M}$, AKTP$^{Null}$, and AKTP$^{M/LOH}$ tumor cells, indicating that the p53 status is important for dictating malignant cell characteristics, except for the proliferation rate (Supplementary Fig. 6).

**Metastasis-initiation by *Trp53* GOF mutation with LOH.** We further examined the efficiency of metastatic foci development by the *Trp53* genotype of AKTP cells using a small number of single cells by transplantation of $3 \times 10^2$ trypsin-dissociated cells to the spleen (Fig. 6a). An intensive histological analysis of serial sectioning revealed small metastatic foci in the liver of three out of four mice that were transplanted with AKTP$^{M/LOH}$ cells (Fig. 6b, c). We confirmed Wnt signaling activation by the Sox17 expression and proliferation of metastasized cells by Ki67 immunostaining (Fig. 6d). In contrast, such metastatic foci were not found in the livers of mice transplanted with other *Trp53* genotype AKTP cells (Fig. 6c). These results support the hypothesis that the combination of a *Trp53* GOF mutation and loss of wild-type *Trp53* promotes metastasis development through the acquisition of survival and clonal expansion properties of disseminated dormant cells.

**Activated pathways by *Trp53* GOF mutation with LOH.** We finally performed RNA sequencing of AKTP cells with the different *Trp53* genotypes as well as adenoma cells carrying simple *Apc* mutation (A cells) as a control. By hierarchical clustering analysis, we extracted three sets of differentially expressed genes (DEGs): AKTP$^{M/LOH}$ and AKTP$^{Null}$ cell-specific, AKTP$^{M/LOH}$ cell-specific, and AKTP$^{Null}$ cell-specific DEGs (Fig. 7a). An ingenuity pathway analysis (IPA) using DEG sets indicated that stem cell pathways were upregulated in both AKTP$^{M/LOH}$ and AKTP$^{Null}$ cells (Fig. 7b). A StemChecker analysis using RNAseq data also indicated the acquisition of stem cell properties in both AKTP$^{M/LOH}$ and AKTP$^{Null}$ cells at a similar level (Supplementary Fig. 7). Importantly, a further IPA analysis indicated that growth factor/MAPK and inflammatory pathways were specifically activated in AKTP$^{M/LOH}$ cells but not in AKTP$^{Null}$ cells (Fig. 7c, d). STRING database search indicated significant interaction among target molecules of these activated upstream regulators, suggesting that growth factor/MAPK and inflammatory pathways generate signaling network in AKTP$^{M/LOH}$ cells (Supplementary Fig. 8).

We thus examined the role of the MAPK pathway in the acquisition of metastatic ability of AKTP$^{M/LOH}$ cells by treatment with the MEK inhibitor trametinib. Notably, trametinib treatment significantly suppressed the cloning efficiency of single-dissociated AKTP$^{M/LOH}$ cells (Fig. 7e). Moreover, the proliferation of AKTP$^{M/LOH}$ cells after seeding $5 \times 10^2$ single-dissociated cells into a 96-well plate was significantly suppressed by treatment with trametinib (Fig. 7f).

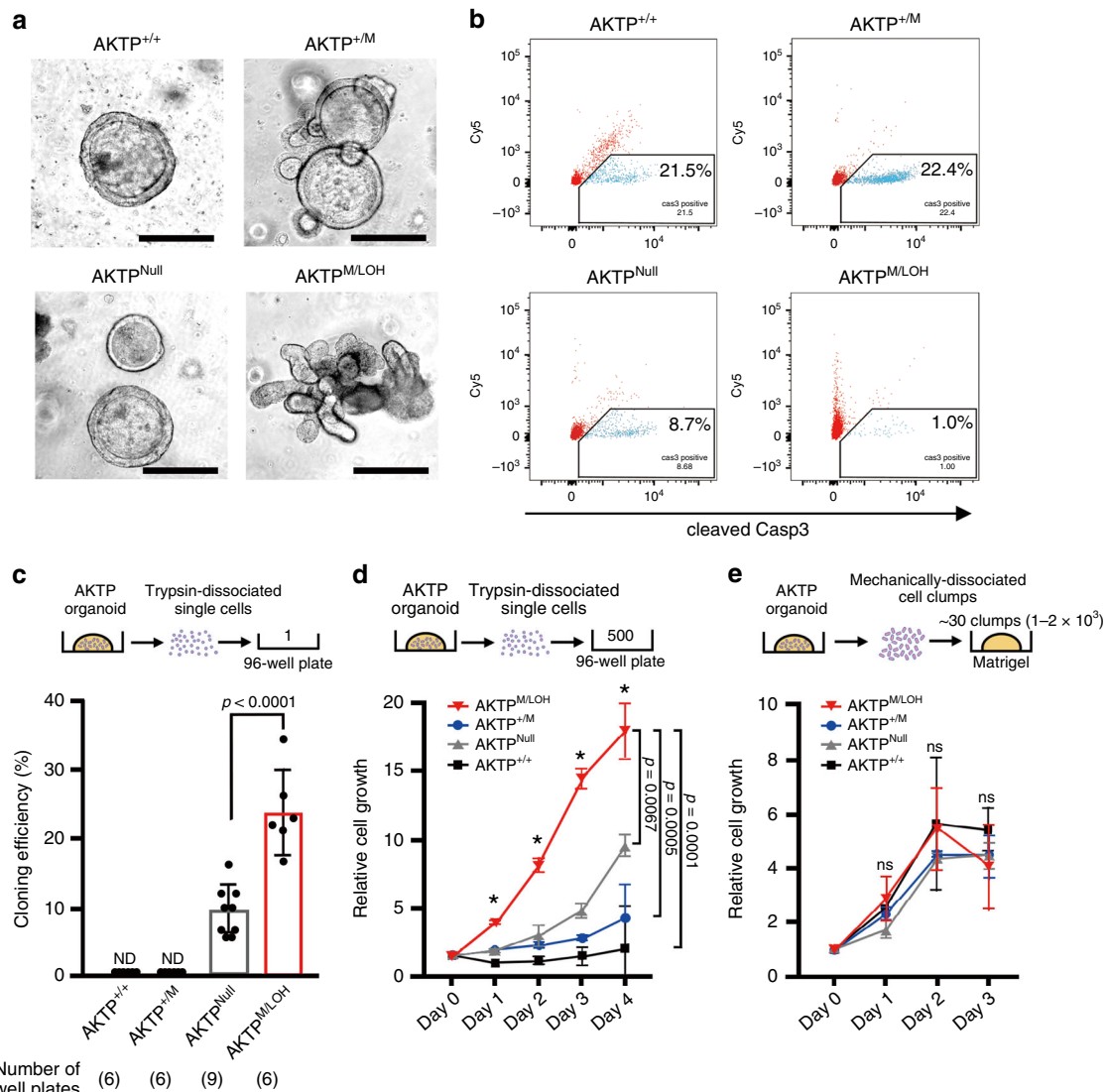

**Fig. 4 Increased survival of dissociated tumor cells by *Trp53* GOF mutation with LOH. a** Representative photographs of four AKTP organoid lines with different *Trp53* genotypes (AKTP[+/+], AKTP[+/M], AKTP[Null], and AKTP[M/LOH]). Bars, 500 μm. Note that AKTP[M/LOH] cells formed a complex glandular structure, while other organoids showed a round cystic structure. The images are representative of six independent cultures. **b** A flow cytometry analysis for cleaved caspase 3 (Casp3) of trypsin-dissociated cells of the four different *Trp53* genotypes. Gating strategy is shown in Supplementary Fig. 5. The apoptosis rate is indicated in each graph. **c** Cloning efficiencies examined by limiting dilution of trypsin-dissociated single cells in 96-well plates are shown as a bar graph. The numbers of 96-well plates used for each genotype analysis are indicated in parentheses. Cloning efficiency was calculated in each 96-well plate, and the mean cloning efficiency was calculated (n = 6–9 biologically independent samples). Gray-line bar, AKTP[Null] cells; red-line bar, AKTP[M/LOH] cells; and ND, not detected. **d, e** The results of a cell proliferation assay of trypsin-dissociated single AKTP cells (500 cells/well) (n = 3 biologically independent samples) (**d**) and mechanically dissociated AKTP cell clumps (~30 clumps containing 1–2 × 10³ cells/Matrigel) (n = 3 biologically independent samples) (**e**). Red lines with triangles, AKTP[M/LOH] cells; blue lines with circles, AKTP[+/M] cells; gray lines with triangles, AKTP[Null] cells; and black lines with squares, AKTP[+/+] cells. The relative cell growth compared to the day 0 level is indicated. Data are presented as mean ± s.d. The data in **c** were analyzed by two-sided unpaired *t*-test, *p* values are provided; ND, not detected. The data in **d, e** were analyzed by ANOVA test, *p < 0.0001; ns, not significant. Data in **d** were also analyzed by Tukey test, *p* values at day 4 are provided. Source data are provided as a Source data file.

Taken together, these results indicate that the loss of wild-type p53 causes the upregulation of the stem cell signature, while the cooperation of the mutant p53 expression and *Trp53* LOH further activates the growth factor/MAPK and inflammatory pathway, which may promote metastasis through the acquisition of single cell survival and clonal expansion ability (Fig. 8).

## Discussion

A recent genome analysis indicated that major driver mutations, including *TP53*, accumulate in the primary tumor cells at the early stage of tumorigenesis[31], but the molecular mechanism

underlying metastasis has not been genetically explained yet. The majority of *TP53* GOF mutations are followed by loss of wild-type *TP53* by LOH, and the possible mechanism underlying *TP53* LOH for tumorigenesis was closely examined through cell line studies[32]. However, exactly when *TP53* LOH is induced during tumorigenesis and what advantages it provides to cancer cells for malignant progression remain unclear.

In the present study, we showed that cancer cells that lost wild-type *Trp53* by LOH were selectively enriched in the metastatic foci when cancer cells carrying a heterozygous *Trp53* mutation were transplanted in the primary site, indicating that

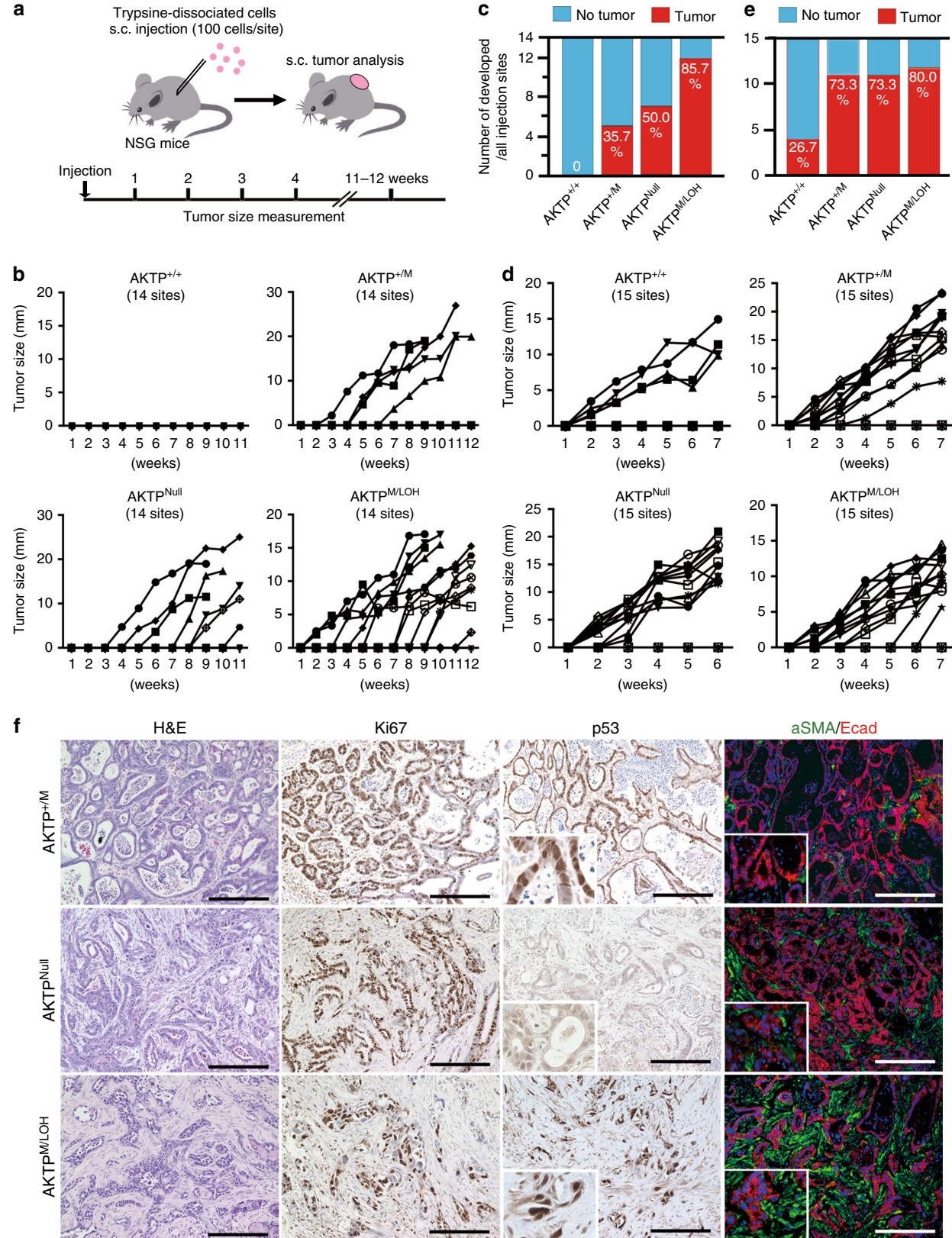

the loss of wild-type *Trp53* by LOH is required for metastasis. Furthermore, in the in vitro analyses, we showed that *Trp53* LOH is not required for the survival or proliferation of cancer cells when they are maintained as tumor glands. However, *Trp53* LOH is critical for the survival when cells are dissociated into single cells. These results indicate that *Trp53* LOH is an

important genetic alteration step for the survival of disseminated cells during metastasis.

The next question is whether or not GOF mutant p53 is required for *Trp53* LOH-associated metastasis. The loss of wild-type p53 causes the protection of dissociated tumor cells from apoptosis by the loss of p53's functions against stress responses.

**Fig. 5 Enhanced tumor-initiating ability by *Trp53* GOF mutation with LOH. a** Strategy of subcutaneous (s.c.) tumor development in NSG mice. **b** The s.c. tumor size changes in each site injected with $1 \times 10^2$ trypsin-dissociated cells of the indicated genotypes are shown as line graphs. The total number of injected sites ($n = 14$) is indicated in each graph. Each line with closed or open circles, squares, diamonds, or triangles indicates the kinetic tumor size development of an individual tumor. **c** The total number of developed tumors (y-axis) at the 14 injected sites is shown by red bars. Blue bars indicate number of sites where tumors did not develop. The ratio (%) of tumor development is indicated inside each bar. **d** The s.c. tumor size changes in each site injected with $5 \times 10^3$ mechanically dissociated cells of the indicated genotypes are shown as line graphs. The total number of injected sites ($n = 15$) is indicated in each graph. Each line with closed or open circles, squares, diamonds, or triangles indicates the kinetic tumor size development of an individual tumor. **e** The total number of developed tumors (y-axis) at the 15 injected sites is shown. The ratio (%) is indicated inside each bar. **f** Representative histology photographs of s.c. tumors that developed from $1 \times 10^2$ trypsin-dissociated cells. Cell genotypes are indicated in the left. H&E and immunohistochemistry for Ki67, p53, and double fluorescent immunohistochemistry for α-smooth muscle active (αSMA) (green)/E-cadherin (Ecad) (red) are shown. The insets show enlarged images. Bars, 200 μm. Source data are provided as a Source data file.

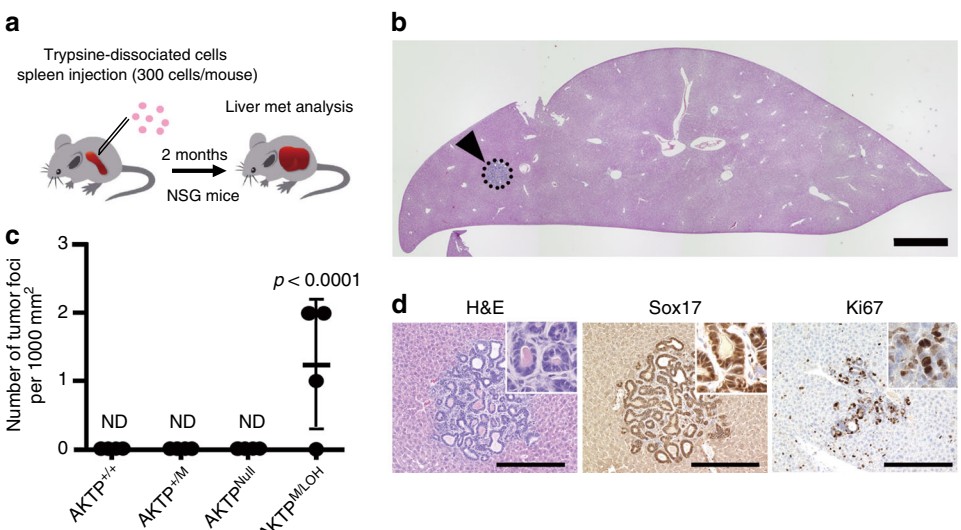

**Fig. 6 Enhanced metastasis-initiating ability of the *Trp53* GOF/LOH tumor cells. a** The strategy of metastasis tumor development in NSG mouse liver. A small number of trypsin-dissociated cells ($3 \times 10^2$ cells) were injected into the spleen. The liver metastatic foci were examined by an intensive histological analysis. **b** Representative histology section of the whole liver lobe of a mouse injected with AKTP$^{M/LOH}$ cells (H&E) is shown. The arrowhead indicates a micrometastatic lesion consisting of AKTP$^{M/LOH}$ cells. Bar, 1 mm. The image is a representative of three independent mice that showed liver metastasis. **c** The numbers of metastatic tumor foci found in the liver in each mouse are shown with closed dots. Data are presented as mean ± s.d., $n = 4$ biologically independent animals. Wilcoxon's nonparametric analysis was used to calculate statistical difference. $p$ value vs. AKTP$^{+/+}$, AKTP$^{+/M}$, and AKTP$^{Null}$ cells is provided; ND, not detected. **d** Representative histology of the serial sections of an AKTP$^{M/LOH}$ cell metastatic lesion (indicated lesion by dotted circle in **b**); H&E and immunohistochemistry for Sox17 and Ki67 (left to right). Bars, 200 μm. The images are representative of three independent experiments. Source data are provided as a Source data file.

However, the present results indicate that the survival rate of enzymatically dissociated cells is significantly higher in AKTP$^{M/LOH}$ cells than in AKTP$^{Null}$ cells. Furthermore, in vivo tumorigenesis from a small number of dissociated single cells is more efficient in AKTP$^{M/LOH}$ cells than in AKTP$^{Null}$ cells. These results indicate that GOF mutant p53 plays a role in the survival and proliferation of dormant cells in addition to the loss of wild-type p53; the combination of a GOF mutation and LOH is thus important for metastasis. However, it has been reported recently that the *Trp53$^{null}$* mutation and the combination of *Trp53$^{R270H}$* GOF/LOH mutations have similar effects on metastasis of intestinal tumors[33]. In this report, tumor phenotypes were examined using mice carrying *Apc*, *Kras*, and *Trp53* (AKP) mutations, and metastasis was found in about 7–8% of both *Trp53$^{null}$* and *Trp53$^{R270H/LOH}$* mice, suggesting the need for additional genetic alterations in order to induce metastasis formation. It is, therefore, possible that the *Trp53* null mutation is sufficient for inducing metastasis without a GOF mutation of *Trp53* when specific genetic alterations other than AKTP mutations are accumulated. However, in line with the present results, it has been shown that a different type of GOF mutant *Trp53*, R248Q, with loss of wild-type *Trp53* induces an advanced

malignant phenotype in an inflammation-associated colon cancer model when compared with *Trp53* null mice[34]. Taken together, these present and previous findings strongly suggest that GOF mutant p53 proteins may be useful therapeutic targets, regardless of the mutation positions.

An expression analysis showed that the stem cell signature is upregulated in both AKTP$^{M/LOH}$ and AKTP$^{Null}$ cells at similar levels, which may contribute to the tumor-initiating ability of cancer cells, although it is not sufficient for the induction of metastasis. However, the inflammation and growth factor/MAPK pathways are strongly activated in AKTP$^{M/LOH}$ but not in AKTP$^{Null}$ cells (Fig. 8). Inflammatory responses contribute to the generation of a fibrotic microenvironment in tumor tissue, and CRC with fibrotic stroma is classified as consensus molecular subtype (CMS)4, which is associated with a poor prognosis[28,35]. Consistently, we found a malignant histology of AKTP$^{M/LOH}$ cells in the metastatic tumors showing a rich fibrotic micro-environment (Figs. 1c, 5f). Furthermore, the activation of growth factor signaling, including that of EGF, PDGF, and VEGF, has also been reported in cancer cells carrying *TP53* GOF mutations[19,21,23]. It is possible that signaling from a fibrotic microenvironment along with the cell-intrinsic activation of these

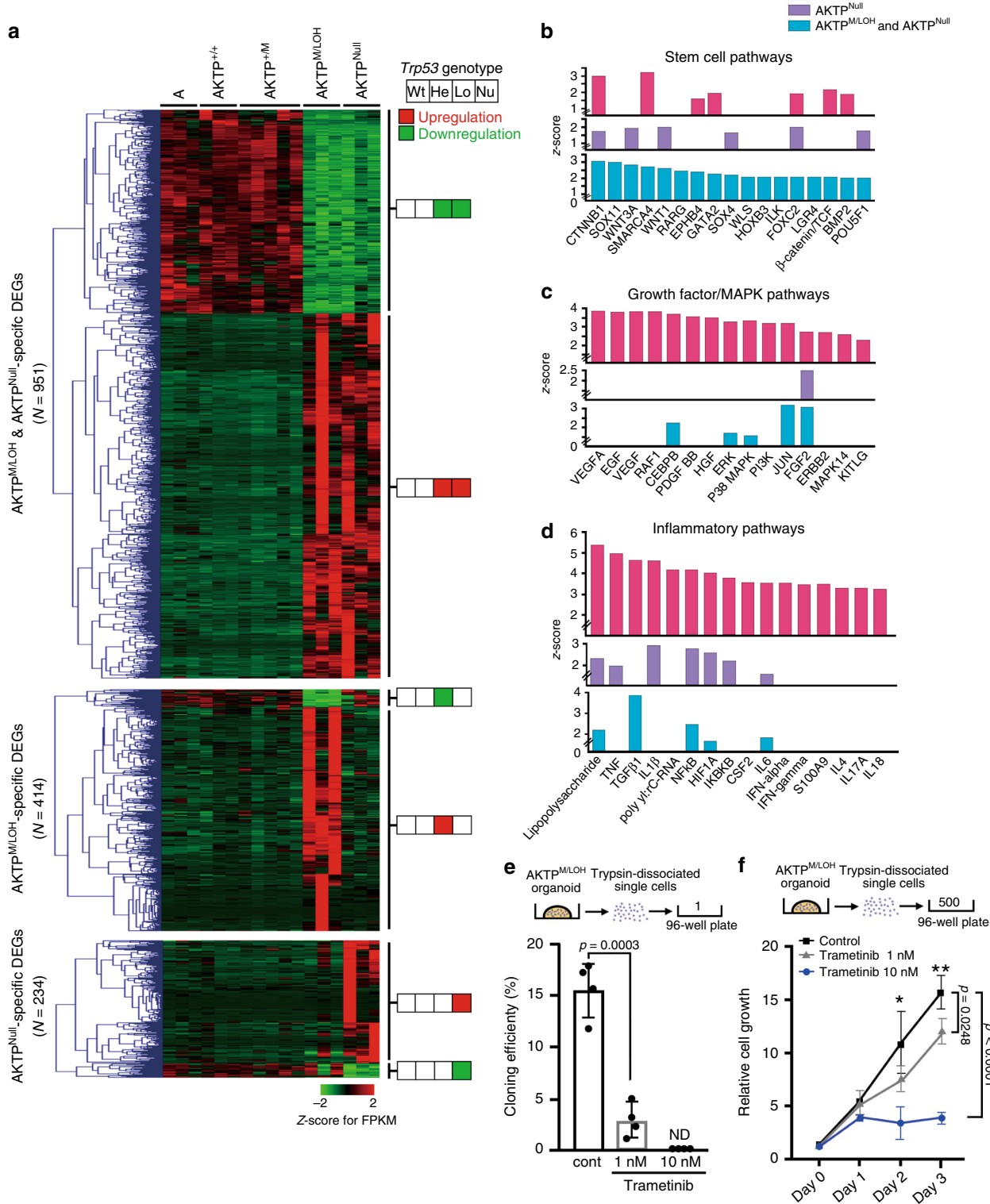

growth factor pathways accelerate the acquisition of a tumor-initiating ability among CRC cells that express GOF mutant p53 and loss of wild-type p53.

We extracted AKTP[M/LOH]-specific DEGs that are not upregulated or downregulated in AKTP[+/M] cells, indicating that the transcription of these genes requires the loss of wild-type p53 function. We previously found that wild-type p53 suppresses the nuclear accumulation of stabilized mutant p53[13,14]. In the present study, we confirmed that the nuclear accumulation ratio of p53

was significantly higher in AKTP[M/LOH] cells than in AKTP[+/M] cells (Figs. 1f, g, and 2d, e). Accordingly, it is possible that the loss of wild-type TP53/Trp53 by LOH in the mutant TP53/Trp53 heterozygous cells leads to the accumulation of mutant p53 in nuclei, which increases the expression of the mutant p53-specific gene set[13]. The loss of the tumor suppressor function of p53 is obviously important for malignant progression. However, the p53 nuclear accumulation may also be an important mechanism through which TP53/Trp53 LOH promotes metastasis.

**Fig. 7 Activation of inflammatory and MAPK pathways by *Trp53* GOF mutation with LOH. a** A hierarchical clustering analysis of AKTP$^{M/LOH}$ and AKTP$^{Null}$-specific DEGs (top), AKTP$^{M/LOH}$-specific DEGs (middle) and AKTP$^{Null}$-specific DEGs (bottom) are shown as fold changes of RPKM compared with the mean. Color scale represents the Z-score log2 intensity. For *Trp53* genotypes, Wt, AKTP$^{+/+}$; He, AKTP$^{+/M}$; Lo, AKTP$^{M/LOH}$; and Nu, AKTP$^{Null}$. The color in the squares on the right of heatmap indicates genes that are upregulated (red) or downregulated (green) compared to A organoid. **b–d** The results of upstream regulator analyses by the Ingenuity Pathway Analysis software program using the AKTP$^{M/LOH}$-specific DEGs (magenta bars), AKTP$^{Null}$-specific DEGs (purple bars), and AKTP$^{M/LOH}$ & AKTP$^{Null}$-specific DEGs (blue bars) mentioned in (**a**) are shown in bar graphs. Upstream regulators for the stem cell pathways with Z-scores > 2.0 in AKTP$^{M/LOH}$ & AKTP$^{Null}$-specific DEGs (**b**), the growth factor/MAPK pathway with Z-scores > 2.0 (**c**) and the inflammatory pathway with Z scores > 3.0 in AKTP$^{M/LOH}$-specific DEGs (**d**) are extracted and pathways with Z-scores > 1.5 are indicated as bars. **e** Cloning efficiencies of trypsin-dissociated AKTP$^{M/LOH}$ cells in the presence or absence of trametinib examined by limiting dilution in 96-well plates are shown ($n = 4$ biologically independent samples). **f** The results of a cell proliferation assay of trypsin-dissociated single AKTP$^{M/LOH}$ cells (500 cells/ well) in the presence or absence of trametinib (mean ± s.d.) ($n = 4$ biologically independent samples). Data are presented as mean ± s.d. The data in **e** were analyzed by two-sided unpaired *t*-test, *p* value is provided; ND, not detected. The data in **f** were analyzed by ANOVA test, *$p = 0.0994$ at day 2, and **$p = 0.0012$ at day 3. The data in **f** were also analyzed by Tukey test, *p* values at day 3 are provided. Source data are provided as a Source data file.

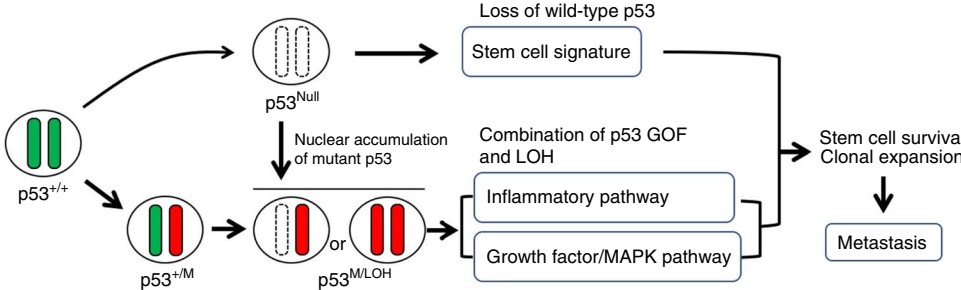

**Fig. 8 A schematic illustration of the for *TP53/Trp53* mutation and metastasis.** Loss of wild-type p53 increases stemness, while combination of p53 GOF mutation with LOH activates inflammatory and MAPK pathway. Corporation of these pathways may cause acquisition of single-cell survival and clonal expansion abilities, which promotes metastasis.

In the present study, we found that AKTFP$^{+/M}$ cells, in which the *Fbxw7* gene is disrupted, survived and expanded from dissociated single cells without the loss of wild-type *Trp53*. Fbxw7 is a ubiquitin ligase that targets oncogenic proteins, including c-Myc[36]. It has been reported that mutant p53 contributes to tumorigenesis by allowing the overexpression of c-Myc through the downregulation of Fbxw7[37]. Thus, the suppression of the Fbxw7 function causes the stabilization of c-Myc without a *Trp53* mutation, which may contribute to the cell survival and proliferation. According to genome research results, certain populations of cancer cells retain intact *TP53*[4]. Therefore, it is possible that *Fbxw7* disruption induces the malignant progression of CRCs that carry intact *TP53*.

Using mouse intestinal tumor cells that carry quadruple driver mutations for *Apc*, *Kras*, *Tgfbr2*, and *Trp53*, we showed that cells remained sensitive to anoikis if their *Trp53* status was heterozygous, and a loss of wild-type *Trp53* by LOH is necessary for the cells to survive in a dormant state and proliferate toward metastasis. We further showed that the inflammation and MAPK pathways were activated by the combination of a *Trp53* GOF mutation with LOH, which may contribute to the acquisition of a malignant phenotype among CRC cells. These results suggest that the inhibition of the mutant p53 function or suppression of *TP53* LOH will be effective for preventing CRC metastasis.

## Methods

**Tumor organoid cell lines**. We previously established AKTP$^{+/M}$, ATP$^{+/M}$, AKP$^{+/M}$, AKF, and AKTFP$^{+/M}$ organoid cell lines from mouse intestinal tumors carrying mutations in the combinations of *Apc$^{\Delta716}$ Kras$^{+/G12D}$ Tgfbr2$^{-/-}$ Trp53$^{+/R270H}$, Apc$^{\Delta716}$ Tgfbr2$^{-/-}$ Trp53$^{+/R270H}$, Apc$^{\Delta716}$ Kras$^{+/G12D}$ Trp53$^{+/R270H}$, Apc$^{\Delta716}$ Kras$^{+/G12D}$ Fbxw7$^{-/-}$* and *Apc$^{\Delta716}$ Kras$^{+/G12D}$ Tgfbr2$^{-/-}$ Fbxw7$^{-/-}$ Trp53$^{+/R270H}$*, respectively[26]. These organoid cells carry heterozygous *Trp53* R270H mutations. AKTP$^{Null}$ cells were generated by the disruption of *Trp53* in AKT (AKTP$^{+/+}$) cells. In brief, *Trp53* CRISPR/Cas9 plasmid (Santa Cruz) was transfected to AKT cells by Lipofectamine LTX, and *Trp53* disrupted cells were selected by the addition of 10 μM Nutlin-3 to the culture medium.

These organoid cells were cultured in growth factor-reduced (GFR)-Matrigel (Corning) or culture dish with Advanced DMEM/F-12 medium (Gibco) supplemented with 10 mM HEPES, 2 mM Glutamax, 1×B27, 1×N2 (Invitrogen), 100 ng/ml murine Noggin (Peprotech) and 1 μM N-acethylcysteine (Sigma).

For the passage of the organoids, organoid cells were recovered from Matrigel by a Cell Recovery Solution (Corning). They were then roughly broken up by pipetting (mechanical dissociation) using 1-ml tips, split and passaged into fresh Matrigel. For the enzymatic dissociation of organoids to single cells, the recovered organoids were treated with 0.25% trypsin at 37 °C for 5 min, and cells were then filtrated using a 35-μm mesh cell strainer (FALCON).

The organoids were immunostained using antibodies for p53 (Leica Biosystems) at 1:200 and E-cadherin (R&D system) at 1:100, and Alexa Fluor 594- or Alexa Fluor 488-conjugated antibodies (Molecular Probes). The immunostained organoids were examined using confocal microscope, Leica TCS SP8 (Leica Microsystems), and the number of cells with p53 nuclear accumulation was scored and the ratio was calculated as nuclear p53 index.

**Mouse experiments**. Female NOD/Shi-*scid Il2rg*−/− (NSG) mice between 6 and 7 weeks of age were purchased (Charles River) for use in transplantation experiments. For the liver metastasis experiments, mechanically dissociated AKTP$^{+/M}$ organoid cells ($1 \times 10^5$ cells/mouse) were injected into mouse spleen, and liver metastases were examined at 4 weeks after the injection ($n = 5$). For s.c. tumor development, trypsin-dissociated organoid cells ($1 \times 10^2$ cells/site) or mechanically dissociated organoid cells ($5 \times 10^3$ cells/site) were injected s.c. (4–5 sites/mouse, and 4 mice each for different cell preparations and genotypes), and the tumor sizes (major diameter, mm) were measured every week until 12 weeks after injection. For in vivo tumor-initiation experiments, trypsin-dissociated organoid cells ($3 \times 10^2$ cells/mouse) were injected into the spleen ($n = 4$), and then the spleen was removed immediately after injection. At 8 weeks after the injection, all liver lobules were collected and processed for histology. Ten paraffin sections (80-μm intervals each) were prepared by microtome and examined. The metastasis frequency was calculated as the number of foci/1000 mm$^2$.

The mice were housed in a 12-h light/dark cycle at 23 °C ± 2 °C room temperature with relative humidity of 50 ± 20%, and given ad-libitum access to food and water for the duration of the study. Mice were housed in specific-pathogen-free (SPF) facility of Kanazawa University, Japan, and cared for in accordance with Fundamental Guidelines for Proper Conduct of Animal Experiment and Related Activities in Academic Research Institutions under the jurisdiction of the Ministry of Education, Culture, Sports, Science and Technology of Japan. All animal experiments were carried out according to the protocol approved by the Committee on Animal Experimentation of Kanazawa University, Japan.

**Histology and immunohistochemistry**. The spleen tumors, liver metastasis tumors, and s.c. tumors were fixated in 4% paraformaldehyde, paraffin-embedded, and sectioned at 4-μm thickness. The sections were stained with H&E or Masson Trichrome. For immunohistochemistry, antibodies against αSMA (Sigma) at 1:800, E-cadherin (R&D Systems) at 1:100, p53 (CM5) (Leica Biosystems) at 1:200, Ki67 (Abcam) at 1:1000 and Sox17 (R&D Systems) at 1:100 were used as the primary antibody. Staining signals were visualized using the Vectastain Elite Kit (Vector Laboratories). For fluorescent immunohistochemistry, Alexa Fluor 594- or Alexa Fluor 488-conjugated antibodies (Molecular Probes) were used as the secondary antibody. The numbers of p53 nuclear-accumulated cells in the liver metastasis foci were scored in 5 microscopic fields, and the ratio was calculated as the mean number of positive cells per tumor gland.

**Proliferation analysis**. The number of Ki67 positive tumor cells and total tumor cells were counted in 5 independent microscopic fields (×200) for 3 independent tumors ($n = 15$ for each genotype), and Ki67 labeling indices were calculated.

**Immunoblotting**. Organoid cells were lysed in lysis buffer, and protein samples were separated in 10% SDS-polyacrylamide gel. Antibodies for p53 (Cell Signaling Technology) was used. An anti-β-actin antibody was used as the internal control. The ECL detection system (GE Healthcare) was used to detect the signals. Band intensities were measured using ImageJ application.

***Trp53* LOH analyses**. To examine the *Trp53* LOH, tumor cells were isolated from eight spleen tumors and eight liver tumors developed in six mice using LMD7000 laser microdissection (Leica Microsystems), and genomic DNA was purified using a NucleoSpin Tissue XS (Macherey-Nagel). Organoid cell-derived genomic DNA was used as a control. To detect the wild-type *Trp53* and mutant *Trp53*R270H, genomic DNA was amplified by PCR using a primer set for *Trp53* (p53F1 and p53R1). Band intensities of the mutant *Trp53*-specific band (330 bp) and wild-type *Trp53*-specific band (290 bp) were measured using ImageJ, and the WT/Mut *Trp53* ratio was calculated.

**Cell culture experiments**. For single-cell subcloning experiments, trypsin-dissociated single cells were plated on a collagen-coated dish or 96-well plates (1 cell/well) and cultured in advanced DMEM/F-12 medium supplemented with inhibitors for ROCK, 10 μM Y-27632 (Wako), GSK-3, 5 μM CHIR99021 (R&D) and TGF-β, 5 μM A8301 (Sigma). Genomic DNA was extracted from the subclones ($n = 5$–17 for each genotype), and genomic PCR for *Trp53* was performed to examine the LOH status. The cloning efficiency was examined by scoring the number of wells containing proliferated cells in the 96-well plates and calculating the ratios.

For the organoid formation assay, $2.5 \times 10^2$ trypsin-dissociated single cells were cultured in 500 μl of collagen type I gel (Nitta gelatin), or 30 cell clumps including $1$–$2 \times 10^2$ cells were cultured in 30 μl of GFR-Matrigel. The numbers of developed organoids were counted at 14 days. For cell proliferation experiments, trypsin-dissociated single cells were plated in 96-well plates ($5 \times 10^2$ cells/well), or $1 \times 10^3$ cells were cultured in 20 μl of GFR-Matrigel. The cell growth was examined by a CellTiter-Glo Luminescent Cell Viability Assay (Promega). All cell culture experiments were repeated three times unless otherwise noted. To inhibit MEK signaling, cells were treated with trametinib (ChemScene) at 1 or 10 nM in the medium.

For the soft agar colony formation assay, $1$–$2 \times 10^3$ cells were mixed in 0.4% agar, seeded into 6-well plates, and cultured for 14 days. The cells were stained with Giemsa Stain Solution (Wako), and the numbers of colonies were counted under a dissecting microscope.

**Flow cytometry analyses**. The trypsin-dissociated single cells were prepared from organoids and fixed with 4% paraformaldehyde for 10 min. After being permeabilized in cold methanol, the cells were stained with Alexa Fluor 488-conjugated anti-cleaved caspase 3 antibody (Cell Signaling Technology) at 1:50 at room temperature for one hour. After washing, apoptotic cells were analyzed using a FACSCanto II (Becton Dickinson).

**DNA sequencing**. Genomic DNA was extracted from organoid cells. The genomic DNA fragment encompassing exons 7 and 8 of *Trp53* was amplified by PCR using a primer set (p53F2 and p53R2). The DNA sequences around codon 270 of *Trp53* was examined by direct sequencing using the forward primer. Genomic DNA from CMT93 cells (ATCC) was used as the *Trp53* wild-type control.

**CGH analyses**. Genomic DNA was extracted and purified from the AKTP+/M and AKTPM/LOH cells, and the whole-genome CGH array was performed by a SurePrint G3 Mouse Genome CGH microarray 1 × 1 M (Agilent).

**Genotyping *Trp53* in cell lines**. Genotyping of *Trp53* in organoid cell lines was performed using a TaqMan SNP genotyping assay (Applied Biosystems). The custom TaqMan probes for wild-type *Trp53* (VIC dye) and mutant *Trp53* R270H (FAM dye) were designed in exon 8, including R270 codon of mouse *Trp53*. The sequences of TaqMan probe and primers (p53F3 and p53R3) are provided below. The PCR results were analyzed by an allele discrimination/SNP analysis system in Mx3000P (Stratagene).

Primer sequences used in the study:

```
p53F1, agcctgcctagcttcctcagg;
p53R1, cttggagacatagccacactg;
p53F2, agactccaggtaggaaggcgcgtggta;
p53R2, cttggtcccgcctgcgtacctctcttt;
p53F3, ccggatagtgggaaccttctg;
p53R3, tcttctgtacggcggtctct.
```

**TaqMan probe used for *Trp53* genotyping**. TaqMan MGB probe to detect wild-type *Trp53* (VIC dye): CTTTGAGGTTC**G**TGTTTGT

TaqMan MGB probe to detect mutant *Trp53* R270H (FAM dye): CTTTGAGGTTC**A**TGTTTGT

**RNA sequencing and analyses**. Total RNA was extracted from organoid cells of AKTPM/LOH, and AKTPNull using an RNeasy plus Micro Kit (Qiagen). RNASeq libraries were prepared using a SureSelect Strand Specific RNA Reagent Kit (Agilent Technologies) according to the manufacturer's protocol. Single-end sequencing with 50 cycles, such that adapters at 3'-ends of reads were not included, was performed using an Illumina HiSeq3000 (Illumina). All reads were demultiplexed on the basis of their unique indices using Illumina's bcl2fastq2 (version 2.20). Sequencing data were deposited in the DNA Data Bank of Japan (DDBJ); accession #DRA005647 and #DRA008701). The quality of raw reads was assessed with FastQC (version 0.11.9)[38]; the quality scores were >Q30, which indicated high quality (Supplementary Fig. 9a). In data processing step, read deduplication was skipped because computational removal of read duplicates, especially in low quantity samples can worsen the power for the differential gene expression[39,40]. Clean reads, for which the average quality scores for A, AKTP+/+, AKTP+/M, AKTPM/LOH and AKTPNull samples were greater than Q30, were processed using the Tuxedo protocol[41] with TopHat2 (version 2.1.1)[42] and Cufflinks[43]. To maintain the previous gene expression data of the A, AKTP+/+, and AKTP+/M samples[26] with the splice junction rate in read alignment, reads for each sample were aligned to the mouse reference GRCm38.p4/mm10 using TopHat2 with the following arguments: (1) segment-length 17, (2) segment-mismatches 1 and (3) library-type fr-unstranded. Although the segment-length was adjusted (25 to 17), the mapping rate of AKTPM/LOH and AKTPNull reads with the two values was found to be almost similar (Supplementary Fig. 10). After sequence alignment, the duplication rate of reads was determined by sequence- and mapping-based approaches using RSeQC[44], and the low distribution of uniquely mapped reads from mapping-based QC (11–18%) were observed (Supplementary Fig. 9b), indicating high rates of read duplicates, possibly due to RNA-Seq library preparation with low input amounts. The sequencing data are summarized in Supplementary Table 1. Gene expression quantification was performed using Cufflinks[43], and Reads Per Kilobase of transcript per Million mapped reads (RPKM) was calculated as the expression value. A principal component analysis (PCA) was performed on the basis of RPKM values to examine whether the samples clustered per their genotype differences (Supplementary Fig. 9c). Differential expression analyses between samples with replicates were performed using Cuffdiff, with the cut-off set at $P < 0.01$ and $\geq 1.5$-fold change (DEGs are presented in Supplementary Data 1), shown in the volcano plot of DEGs with statistical significance (Supplementary Fig. 9d). Hierarchical clustering in the gene expression data was analyzed with MeV (http://mev.tm4.org) using Euclidean distance and the complete linkage method. For the upstream regulator analysis, the AKTPM/LOH- and/or AKTPNull-specific gene set was analyzed using the Ingenuity Pathway Analysis software package (Ingenuity Systems; www.ingenuity.com). Pathways with z-scores of >2 (for inflammatory pathways) or >3 (for growth factor/MAPK pathways) and P-values of <0.05 were designated as significantly activated.

**Mutual exclusivity analysis**. Mutual exclusivity of *TP53* mutation and *FBXW7* mutation in the human colorectal cancer TCGA database was examined using cBioPortal (https://www.cbioportal.org).

**StemChecker analysis**. The stemness signature in AKTP+/M, AKTPNull and AKTPM/LOH was analyzed using StemChecker[45], a web-based online tool for examining the stemness signatures in user-defined gene sets based on the curation of 50 published stemness signatures (http://stemchecker.sysbiolab.eu).

**STRING database analysis**. Genes that were predicted to be targeted by 31 upstream regulators (Fig. 7c, d) were selected. For protein-protein interaction network analysis, those genes were searched against STRING database[46] with medium confidence score (>0.4). A pattern arising as a result of interactions between genes was visualized with a heatmap.

**Statistical analyses**. The data were analyzed using a two-sided unpaired *t*-test otherwise mentioned and presented as the means ± standard deviation (s.d.). Statistical analyses for Figs. 4d, e and 7f were performed using a one-way ANOVA, followed by Tukey's post-hoc test. Statistical analyses for Fig. 6c were performed using Wilcoxon's nonparametric analysis. A value of $p < 0.05$ was considered to be statistically significant. Excel (16.23, Microsoft) and Graphpad Prism7 were used for statistical analyses.

**Reporting summary**. Further information on research design is available in the Nature Research Reporting Summary linked to this article.

## Data availability

The RNA sequencing data have been deposited in the DNA Data Bank of Japan (DDBJ) under the accession numbers: DRA005647 and DRA008701. The source data underlying Figs. 1d–g, 2b, c, e, f, 3a, c, 4c–e, 5b–e, 6c, 7b–f and Supplementary Figs. 3, 4, 6 are provided as a Source Data file. cBioPortal (https://www.cbioportal.org) has been used to examine mutual exclusivity of genetic alterations in human CRC. STRING database (https://string-db.org) has been used for interaction analysis of pathways. All other data supporting the findings of this study are available within the article and its Supplementary information files and from the corresponding author upon reasonable request.

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

## Acknowledgements

We thank Seung-il Yoo (Theragen Etex Bio Institute) for the assistance of RNA-Seq quality control metrics. We thank Yoshie Jomen, Ayako Tsuda, and Manami Watanabe for their technical assistance. This work was supported by AMED (19ck0106259h0003) from the Japan Agency for Medical Research and Development, Japan; and Grants-in-Aid for Scientific Research (A) (18H04030) and (C) (17K07162), and Grants-in-Aid for Scientific Research on Innovative Areas-Platforms for Advanced Technologies and Research Resources (16H06279) from the Ministry of Education, Culture, Sports, Science and Technology of Japan.

## Author contributions

M.N., E.S., and H.O. performed the experiments and analyzed the results. C.P.H. and S.J.K. performed genetic and bioinformatic analyses. M.O. designed the project and prepared the paper.

## Competing interests

The authors declare no competing interests.
