## [Peer Review File · Nature Communications]

Reviewers' comments:

Reviewer #1 (Remarks to the Author); expert in p53, CRC, mouse models:

Sporadic human colorectal carcinoma (CRC) exhibits p53 alterations with an incidence of 60% , being in fact the second most common mutation in CRC after constitutive activation of the Wnt pathway. They occur at the transition of large adenoma to invasive carcinoma with an incidence of 60%, but increase to an 80% incidence in liver metastasis. Thus, additional p53 genetic change correlates with progression. The vast majority of p53 alterations in CRC are missense mutations in the DNA binding domain with 6 hotspots including R273H (aka R270H in mouse). Notably, 80% of these missense mutant p53 human CRCs have undergone LOH, thus selected against the remaining wtp53 allele in initially heterozygous p53 mutant malignant glands.

This study from the Oshima lab uses a genetically precisely defined and highly relevant quadruple driver gene series (APC, KRas, Tgfbr2, and p53) of mouse CRC organoids to systematically examine the role of LOH of the retained wtp53 allele in a hotspot missense mutant p53 R270H heterozygous model. The authors use clonogenicity, morphology, growth and anoikis assays in vitro, as well as allogeneic transplantation assays into NSG mice into the spleen and subcutaneous tissue to show that LOH is required for survival of disseminated single cells and their clonal expansion that manifests itself by metastatic tumor initiation in vivo, in short their metastatic ability.

Overall, this is an outstanding study that is well controlled and fully justifies its conclusions, namely that there is a 2-step pathway of how missense mutant p53 drives both tumor initiation and malignant progression: initially heterozygous p53 missense mutant alleles exert a gain-of-function (GOF) role that includes - an albeit incomplete - dominant negative action on the retained wtp53 allele, but subsequently cooperates with wtp53 allele loss (LOH) to greatly accelerate local invasion, survival of dormant single malignant cells and their metastatic seeding ability into distant organs. The authors show that LOH prevents the remaining tumor suppressive arrest/apoptotic stress response of wtp53. Moreover they show that LOH abruptly expands the genomic range of actions of missense mutp53 to enable it to activate EMT programs as well as inflammatory and growth factor/MAPK pathways to drive a fibrotic pro-tumorigenic microenvironment. (Interestingly, a stem cell transcriptome was already activated by wtp53 loss alone). This 2-step cooperative GOF/LOH model is very convincing since it fully explains the clinical observations of mainly seeing missense mutp53/LOH tumors, (> 93% of all human cancers) which have long been at odds with the - at least theoretically - competing notion that missense p53 mutants do not have neomorphic GOF function but solely act via a dominant negative 'poisoning' interaction by forming mixed mutant/wildtype tetramers in heterozygosity. The current study also comes at an important time since the dom-neg model-only idea has recently seen some support in conceptually limited established cancer cell line-based studies (Boettcher et al, Science 2019; Giacomelli et al Nat Genet 2018). The Oshima study now definitively puts this discussion to rest by showing in fact a cooperative dual action in sequence of p53 GOS + LOH in an extended driver function of mutant p53 in epithelial cancers such as CRC.

Moreover, an additional interesting point the paper makes is that it identifies an underlying genetic mechanism of metastasis, i.e. wtp53 LOH in the setting of missense p53. This is important because non-genetic epigenetic mechanisms of metastasis were favored since so many attempts to identify 'metastasis genes ' had failed in the past.

These points should be addressed/corrected before publication in Nature Communications:

In Fig 1e, the authors show that LOH increases the ratio of tumor cells in vivo with strictly nuclear p53 stabilization, rather than a mixed nuclear/cytoplasmic localization. This should be more clearly stated in the legend. Moreover, data should be added to Fig 2 hat shows the percentage of nuclear-only localization versus mixed nuclear/cytoplasmic in in vitro organoids.

Statistics is deficient: Many figures and panels are missing n numbers of samples used. Add in all cases. Add error bars where missing, i.e. Fig 4e, 5b, d, 6d.

Figs 1d and 2c: More than 3 samples each of spleen and liver should be examined. At least 10 samples each should be done. Also, in Fig 2c gel and bar graph data need to be shown as in Fig 1d.

Concerning clear allele designation, the name AKTPLOH does not refer to the presence of the missense mutp53 R270H allele, the focus of the study. Renaming the allele to AKTP M/LOH should be considered.

Figs 4c, d, e: Add clear description of what cells (e.g. trypsin-dissociated single cells, mechanically- dissociated clumps of cells in Matrigel etc) were used for each panel.

Figs 5c, e: The figures are supposed to show tumor incidences with AKTP+/M and AKTPnull of 35.7 and 50%, respectively, and AKTPLOH with 85.7%. However, the y-axis scales (0-14 and 0-15; number of tumor sites, what is the total n number in each?) do not make sense. Please correct to indicate both absolute numbers of sites out of total number of sites, as well as percentage.

Fig 5b, d: The legend needs to point out that each line is the kinetic tumor size development of an individual mice, I assume.

Fig 5f legend: add how many trypsin dissociated cells were seeded.

The authors should also quote an important recent paper on mutp53 in CRC (Schulz-Heddergott et al Cancer Cell 2018)

Explain Stem Checker Analysis in Methods.

Fig 1d legend: should say ..." each lane indicates tumor cells from spleen and liver..." because injected spleen tumors are not metastatic.

Fig 3c: should say" is shown in yellow triangles "

Reviewer #2 (Remarks to the Author); expert in CRC organoids:

The manuscript by Nakayama, et al addresses a somewhat controversial topic of whether LOH at the p53 allele combined with expression of a gain-of-function mutation promotes metastatic activity. The novelty of this manuscript lies in the analysis of the behavior of dissociated single cells versus fragments of dissociated tumors. The data with the dissociated cells supports a concept the LOH is a critical factor in metastasis and the ability of single cells to survive. This paper is an important addition to the discussion of the mechanisms that are manifested to promote metastatic behavior (as opposed to just tumor cell proliferation and growth. The data are generally well but several issues should be addressed:

Major concerns:

1) The results in figure 1d and e lack statistical comparison and it is difficult to determine how many tumors were compared from how many animals. This is important because Fig 1d establishes the argument for LOH. Also, it is not clear to this reviewer why the intensity of the wt p53 band is so much less in the genomic PCRs.

2) Figure 2 shows an alteration in organoid formation in Collagen I and Matrigel for cells with LOH. Is there a difference in growth in soft agar, a more traditional anchorage-independent growth measure?

3) The results presented here seem to conflict with results from Eric Fearon's laboratory recently published in *Lab. Invest* (PMID:31148594). Those studies were performed in AKP rather than AKTP mice, but they did not show a significant difference in tumor formation and metastasis between TP53 null and TP53(R270H). This paper's results must at least be compared with the findings in the present paper.

4) The methods say that the data were analysed by unpaired t-test, but that is certainly not appropriate for Figure 4d where ANOVA with post hoc test should be done or in Figure 6 where the n is only 4 (a non-parametric test should be used. I doubt that anything will change but the appropriate tests should be employed. In Figure 5 I am not clear whether any statistical analysis was employed to compare tumor growth, but this would seem to be needed.

Reviewer #3 (Remarks to the Author); expert in cell death and anoikis:

Review of NCOMMS-19-25919

Loss of wild-type p53 promotes mutant p53-driven metastasis through acquisition of survival and tumor-initiating properties.

In this paper the authors present data showing that, in a mouse model, intestinal tumors with one copy of a gain of function p53 mutation and one WT copy, loss of the WT copy (LOH) was seen to be enriched in metastatic sites. p53 often contains missense mutations at hot spots, and these are often oncogenic through a gain of function. The loss of WT p53 is often observed in human cancers where the second allele has these GOF substitutions. Furthermore, functional studies by the authors and another group have shown that the loss of the WT allele in cells with a GOF mutations results in stabilisation and nuclear localisation of the mutant p53. Thus, there is already a considerable understanding of how LOH of p53 contributes to tumour progression.

In this study the authors use an elegant model in which they have engineered Apc, KRas, TGFBR mutations, all common driver mutation in colorectal cancer, as well as the heterozygous p53 with an R270H substitution. They transplanted tumor organoids into the spleen of mice, and then assessed the genotype of cells found metastasised to the liver. The WT p53 allele was frequently lost in the liver metastases. The previous data showing mutant p53 stabilisation following loss of WT p53 was supported by data here showing that LOH organoids had increased nuclear p53. When they looked for a mechanism, they found that if tumour organoids were dissociated into single cells, only cells with LOH expanded clonally. In contrast, if they were cultured as organoids, cells retained the WT p53 allele. It appears that loss of the WT p53 results in increased resistance to anoikis, which has been shown in a number of studies to increase metastasis of tumours and promote secondary tumors. Thus, the key findings are not surprising and generally follow on from previous work. Importantly, all the effects required the mutant p53, as loss of both p53 alleles did not support metastasis or anoikis resistance.

The authors also undertake gene expression studies by RNAseq on the different p53 genotypes. Interestingly they found that LOH was associated with both increased inflammatory pathway genes, and increased expression of growth factor/MAPK genes. These would certainly fit with the observations in mice, but they haven't really made a functional link in this study.

Overall, the data are convincing and technically sound as far as I can see. The question is how the gene expression data explains the phenotype, and it is not clear if this link is made. The key gene expressions signatures would fit the findings. For example, many of the MAPK signalling changes would promote increased resistance to apoptosis. However, they have not directly made this link,

Minor points.

In some places they often describe data as a percentage, for example in Figure 2d they state the nuclear p53 index. However, it is not clear how many cells or tumor organoids were analysed to get this nuclear accumulation ratio.

RESPONSE TO REVIEWER 1

Overall, this is an outstanding study that is well controlled and fully justifies its conclusions, namely that there is a 2-step pathway of how missense mutant p53 drives both tumor initiation and malignant progression: initially heterozygous p53 missense mutant alleles exert a gain-of-function (GOF) role that includes - an albeit incomplete - dominant negative action on the retained wtp53 allele, but subsequently cooperates with wtp53 allele loss (LOH) to greatly accelerate local invasion, survival of dormant single malignant cells and their metastatic seeding ability into distant organs.

[...] The current study also comes at an important time since the dom-neg model-only idea has recently seen some support in conceptually limited established cancer cell line-based studies (Boettcher et al, Science 2019; Giacomelli et al Nat Genet 2018). The Oshima study now definitively puts this discussion to rest by showing in fact a cooperative dual action in sequence of p53 GOS + LOH in an extended driver function of mutant p53 in epithelial cancers such as CRC. [...] This is important because non-genetic epigenetic mechanisms of metastasis were favored since so many attempts to identify ‘metastasis genes ‘ had failed in the past.

Response: We thank Reviewer 1 for positive and constructive comments on our manuscript. We have responded to the comments by Reviewer 1 as follows.

1. In Fig 1e, the authors show that LOH increases the ratio of tumor cells in vivo with strictly nuclear p53 stabilization, rather than a mixed nuclear/cytoplasmic localization. This should be more clearly stated in the legend. Moreover, data should be added to Fig 2 that shows the percentage of nuclear-only localization versus mixed nuclear/cytoplasmic in in vitro organoids.

Response: As suggested by Reviewer 1, we have now clearly stated the subcellular localization of p53 as nuclear-only but not mixed nuclear/cytoplasmic as follows: “...the ratios of p53 nuclear accumulation without cytoplasmic distribution (nuclear-only accumulation) were higher in the liver metastasized tumor cells than in the primary spleen tumor cells (Fig. 1c, f)” (page 7, lines 5-7). The Fig. 1f legend (originally Fig. 1e) was updated accordingly (page 30).

In addition, we examined the subcellular localizations of p53 in the individual organoid cells and classified them as nuclear-only, mixed nuclear/cytoplasmic and cytoplasmic-only. As a result, the ratio of organoid cells with nuclear-only p53 was significantly higher in AKTP^{M/LOH} (48.0%) than in AKTP^{+M} (6.8%). In contrast, the majority of AKTP^{+M} cells showed nuclear/cytoplasmic (73.0%) or cytoplasmic-only distribution of p53 (11.6%), which were rarely found in AKTP^{M/LOH} cells. We have added these data to Fig. 2e and the text (page 8, line 1-5). The Fig. 2f legend was updated accordingly (page 31).

2. Statistics is deficient: Many figures and panels are missing numbers of samples used. Add in all cases. Add error bars where missing, i.e. Fig 4e, 5b, d, 6d.

Response: We apologize for not including the numbers of samples in Figures. We have now added the number of samples to Fig. 1f (100 tumor glands [>20 cells/gland] in 6 microscopic fields on the histology sections), Fig. 2e (526 and 843 cells from 12 organoids each for AKTP^{+M} and AKTP^{M/LOH} cells, respectively), Fig. 2f (4,500 and 3,000 cells seeded in collagen

gel and Matrigel), and Fig. 4c (number of 96-well plates used for each genotype analysis), Fig. 4d (500 cells/well in 96-well plate), Fig. 4e (approximately 30 clumps containing $1-2 \times 10^3$ cells), and Fig. 5b,d (14 and 15 sites for single-dissociated cell and mechanically-dissociated cells, respectively).

In addition, as suggested, we have added error bars as well as “ns” (not significant) and asterisks (significant) to Fig. 4e and Fig. 6c. To respond to Reviewer 2’s comment, we performed an ANOVA with a post hoc test for Fig. 4d, e and a non-parametric test for Fig. 6c. in the revision. Concerning Fig. 5b and d, these line chart graphs indicated kinetic tumor size changes of “individual” s.c. tumors at all injected sites (14 and 15 sites in 3-4 mice for Fig. 5b and 5d, respectively) and thus could not be examined statistically.

3. Figs 1d and 2c: More than 3 samples each of spleen and liver should be examined. At least 10 samples each should be done. Also, in Fig 2c gel and bar graph data need to be shown as in Fig 1d.

Response: We additionally collected LMD samples and examined the *Trp53* genotypes in eight samples each for spleen and liver tumor lesions for Fig. 1d. Notably, we found that four samples showed the clear loss of wild-type *Trp53* in the liver samples, while LOH was not found in any of the spleen samples. It is possible that the liver tumors without loss of wild-type *Trp53* originated from tumor cell clusters including *Trp53* non-LOH cells that had disseminated from the spleen. We believe that this is a reasonable conclusion because AKTP^{+M} cells can survive and proliferate without LOH if a partial glandular structure is maintained, as indicated in Fig. 4e. We further performed statistical analyses of the band intensities between liver samples with LOH or non-LOH and spleen samples. We have added these data to Fig. 1d, e and the text (page 6, line 21-page 7, line 1).

For Fig. 2c, as suggested by Reviewer 2, we developed 10 subclones of the genotypes AKTP, ATP, AKP and AKTFP and confirmed that all subclones from AKTP, ATP and AKP cells lost wild-type *Trp53* by LOH. However, 6 of the 10 AKTFP-derived subclones still contained the wild-type *Trp53* gene. These results suggest that Fbxw7 disruption can induce malignant progression of CRC cells carrying wild-type p53, as discussed in the text (page 16, last paragraph). We have added new genomic PCR results for all genotypes to Fig. 2c with a bar graph of the band intensities for AKTFP subclones.

4. Concerning clear allele designation, the name AKTPLOH does not refer to the presence of the missense mutp53 R270H allele, the focus of the study. Renaming the allele to AKTP M/LOH should be considered.

Response: As Reviewer 1 suggested, we changed the genotype indication for p53 GOF/LOH mutant cells to AKTP^{M/LOH} instead of AKTP^{LOH} throughout the text and Figures. This indication now correctly presents the p53 genotype status.

5. Figs 4c, d, e: Add clear description of what cells (e.g. trypsin-dissociated single cells, mechanically-dissociated clumps of cells in Matrigel etc) were used for each panel.

Response: To respond to this comment, we have now added schematic drawings that indicate what cell types (trypsin-dissociated single cells or mechanically dissociated cell

clumps) were cultured in 96-well plates or Matrigel in Fig. 4c, d, and e. In addition, we have added explanations about the cells to the text (page 10, line 18, 21-23) and Fig. 4c-e legends (page 32).

6. *Figs 5c, e: The figures are supposed to show tumor incidences with AKTP+/M and AKTPnull of 35.7 and 50%, respectively, and AKTPLOH with 85.7%. However, the y-axis scales (0-14 and 0-15; number of tumor sites, what is the total n number in each?) do not make sense. Please correct to indicate both absolute numbers of sites out of total number of sites, as well as percentage.*

Response: We apologize for our description of the tumor development efficiency being confusing due to our using percentages in the text while showing exact tumor numbers in Fig. 5c and 5e. As suggested by Reviewer 1, we have now indicated the numbers of tumors developed among the total injected sites along with the percentage in the text (page 11, lines 9-11, 13-14, and 17-18). We have also indicated the percentages inside the bars of Fig. 5c and 5e while showing the exact tumor numbers in the y-axes. The Fig. 5c and 5e legends were updated accordingly (page 32 and 33).

7. *Fig 5b, d: The legend needs to point out that each line is the kinetic tumor size development of an individual mice, I assume.*

Response: As the reviewer mentions, the Fig. 5b and 5d indicate the kinetic tumor size development of individual tumors. We have now added this explanation to the legends of Fig. 5b and 5d as follows: “Each line indicates the kinetic tumor size development of an individual tumor” (page 32 and 33).

8. *Fig 5f legend: add how many trypsin dissociated cells were seeded.*

Response: The number of trypsin-dissociated cells for s.c. injection experiments is shown in the schematic drawing of Fig. 5a. We have now added text mentioning the number of transplanted trypsin-dissociated cells to the Fig. 5f legend as follows: “Representative histology of photographs of s.c. tumors that developed from 1×10^2 trypsin-dissociated cells.” (page 33)

9. *The authors should also quote an important recent paper on mutp53 in CRC (Schulz-Heddergott et al Cancer Cell 2018).*

Response: The recent paper (Schulz-Heddergott et al, Cancer Cell, 2018) indicated by Reviewer 1 found that the expression of a different type of GOF mutant p53 (R248Q) with loss of wild-type p53 induces an advanced malignant phenotype in an inflammation-associated colon cancer mouse model compared with p53 null mutant mice. The present and these previous findings strongly suggest that GOF mutant p53 can be an effective therapeutic target, regardless of the mutation positions (e.g. R273H and R248Q). We have now discussed this point in the text (page 15, lines 9-14) with a citation of the indicated reference in #34.

10. *Explain Stem Checker Analysis in Methods.*

Response: StemChecker is a web-based online tool for examining the stemness signatures in user-defined gene sets at <http://stemchecker.sysbiolab.eu>. This protocol was developed by Pint et al. (Nucleic Acid Res, 2015) based on the curation of nearly 50 published stemness signatures. We have added this explanation of StemChecker and its web address to the Supplementary Methods along with a citation of the original article.

11. Fig 1d legend: should say ...” each lane indicates tumor cells from spleen and liver...” because injected spleen tumors are not metastatic.

Response: We have now corrected the sentence to “*Each lane indicates the tumor cells collected from the spleen and liver tumors*” in the Fig. 1d legend (page 30).

12. Fig 3c: should say” is shown in yellow triangles ”

Response: As suggested, we have now corrected the indication for “no template control (NTC)” in Fig. 3c from “yellow dots” to “yellow triangles” in the Fig. 3c legend (page 32, lines 2-3).

RESPONSE TO REVIEWER 2

[...] The novelty of this manuscript lies in the analysis of the behavior of dissociated single cells versus fragments of dissociated tumors. The data with the dissociated cells supports a concept the LOH is a critical factor in metastasis and the ability of single cells to survive. This paper is an important addition to the discussion of the mechanisms that are manifested to promote metastatic behavior (as opposed to just tumor cell proliferation and growth. The data are generally well but several issues should be addressed:

Response: We thank Reviewer 2 for positive and constructive comments on our manuscript. We have responded to the comments by Reviewer 2 as follows.

Major concerns:

1) The results in figure 1d and e lack statistical comparison and it is difficult to determine how many tumors were compared from how many animals. This is important because Fig 1d establishes the argument for LOH. Also, it is not clear to this reviewer why the intensity of the wt p53 band is so much less in the genomic PCRs.

Response: To respond to Reviewer 2's comment, we increased the number of LMD samples and ultimately isolated tumor cells from eight spleen tumors and eight liver tumors developed in six mice in total and performed genomic polymerase chain reaction (PCR). We have now added this information to the Methods section (page 20, lines 10-11).

Notably, we found that 4 of 8 samples (50%) showed loss of wild-type *Trp53* among liver tumors, while LOH was not found in any spleen samples. It is possible that the liver tumors without loss of wild-type *Trp53* originated from tumor cell clusters including *Trp53* non-LOH cells that had disseminated from the spleen. We believe that this is a reasonable conclusion because AKT^{+/M} cells can survive and proliferate without LOH if a partial glandular structure is maintained, as indicated in Fig. 4e. As suggested by Reviewer 2, we performed statistical analyses for band intensities between liver tumors with or without LOH (4 samples each) and spleen tumors (8 samples) in Fig. 1e and confirmed the decrease in wild-type *Trp53* in LOH tumors with statistical significance.

For the statistical analysis of Fig. 1f (formerly Fig. 1e), we compared the mean number of tumor glands in which >80% of cells showed nuclear-only p53 in the 6 microscopic fields each of liver and spleen tumors. As expected, the ratio of p53 nuclear accumulation was significantly higher in the liver tumors than in the spleen tumors. We have now added these results to the text (page 7, lines 7-9) and Fig. 1g. The figure legends were updated accordingly (page 30).

We performed genomic PCR for *Trp53* according to the protocol described in the original *Trp53* R270H mouse paper (Olive et al, Cell, 2004). We confirmed that this PCR system amplifies both wild-type *Trp53* and mutant *Trp53* R270H fragments using template DNAs extracted from *Trp53* +/+, *Trp53* M/M as well as *Trp53* +/M cells. As Reviewer 2 mentioned, the wild-type band intensity is always less when *Trp53* +/M cell DNA is used as a template, although the reason for this is unclear. It is technically difficult to construct a different PCR system to detect a mutant *Trp53* gene that carries a short loxP sequence, so we used this system for the LOH analysis. However, to confirm the band size for mutant and wild-type *Trp53*, we have now added the control PCR results using *Trp53* +/+ and *Trp53* M/M cells in addition to *Trp53* +/M cells to Fig. 2c of the revised manuscript.

2) Figure 2 shows an alteration in organoid formation in Collagen I and Matrigel for cells with LOH. Is there a difference in growth in soft agar, a more traditional anchorage-independent growth measure?

Response: As suggested by Reviewer 2, we performed a soft agar colony formation assay using trypsin-dissociated AKTP^{+M} and AKTP^{M/LOH} cells. However, neither cell line formed colonies in soft agar, indicating that AKTP^{M/LOH} cells do not yet have an anchorage-independent growth ability. In other words, signaling from the extracellular matrix (ECM) is required for clonal expansion of AKTP^{M/LOH} cells. We have now added the results of the soft agar colony formation assay to the text (page 8, lines 14-17) without showing any negative data. The Methods section was updated accordingly (page 21, lines 13-15).

3) The results presented here seem to conflict with results from Eric Fearon's laboratory recently published in Lab. Invest (PMID:31148594). Those studies were performed in AKP rather than AKTP mice, but they did not show a significant difference in tumor formation and metastasis between TP53 null and TP53(R270H). This paper's results must at least be compared with the findings in the present paper.

Response: In the paper mentioned by Reviewer 2, the authors showed that *Trp53*^{null} mice and *Trp53*^{R270H/-} mice showed similar malignant phenotypes of intestinal tumors and metastasis. Based on these results, the authors concluded that R270H mutant p53 does not manifest definite GOF biological effects in colorectal cancer. However, the genotypes of the mouse models used in the study differed from those used in the present study (AKP triple mutations vs. AKTP quadruple mutations). Furthermore, metastasis in distant organs in AKP mice was found in only about 7%-8% of cases (1 out of 13-14 mice), suggesting the need for additional genetic alterations in order to induce metastasis formation. It is therefore possible that the p53 null mutation is sufficient for inducing metastasis without a GOF mutation of p53 when specific genetic alterations other than AKTP are introduced, and a GOF mutation may not be important in such cases. We have discussed this possibility in the Discussion section (page 15, lines 29). The indicated paper was added as reference #33.

4) The methods say that the data were analysed by unpaired t-test, but that is certainly not appropriate for Figure 4d where ANOVA with post hoc test should be done or in Figure 6 where the n is only 4 (a non-parametric test should be used. I doubt that anything will change but the appropriate tests should be employed. In Figure 5 I am not clear whether any statistical analysis was employed to compare tumor growth, but this would seem to be needed.

Response: As suggested, we performed an ANOVA for the data in Fig. 4d and 4e and found a significant difference ($p < 0.05$) in variance among the 4 genotypes when cells were trypsin-dissociated (Fig. 4d) but not when they were mechanically dissociated (Fig. 4e). We next performed a post-hoc analysis using Tukey's test for the data for Fig. 4d at day 4 and found that the AKTP^{M/LOH} cell growth was significantly greater than that of AKTP^{+/+}, AKTP^{+M} and AKTP^{Null} cells. We have now added these statistical analysis results to Fig. 4d and 4e. The Methods (page 23, line 20-21) and Figure legend (page 32, lines 12-13) were also updated accordingly.

For Fig. 6c, we performed a Wilcoxon's nonparametric analysis and confirmed the significant difference in the liver metastasis formation after the injection of 300 single-dissociated cells between AKTP^{M/LOH} cells and AKTP^{+/+}, AKTP^{+/M} or AKTP^{Null} cells. We have now updated the Methods (page 23, lines 21-22) and Fig. 6c legend (page 33).

The line chart graphs shown in Fig. 5b and 5d indicate the kinetic tumor size development of individual tumors. As seen in these Figures, the week of the tumor onset varies among individual tumors. We therefore compared the incidence of s.c. tumor development within 12 weeks (Fig. 5b) or 7 weeks (Fig. 5d) among 14-15 injection sites of 3-5 mice, and the exact numbers of tumors and percentages are shown in Fig. 5c and 5e. While it is difficult to examine the results statistically, we believe that the results strongly suggest the increased clonal expansion ability of AKTP^{M/LOH} cells compared with AKTP^{+/+}, AKTP^{+/M} and AKTP^{Null} cells. We hope that Reviewer 2 agrees with our thoughts.

RESPONSE TO REVIEWER 3

[...] In this study the authors use an elegant model in which they have engineered Apc, KRas, TGFBR mutations, all common driver mutation in colorectal cancer, as well as the heterozygous p53 with an R270H substitution. [...] When they looked for a mechanism, they found that if tumour organoids were dissociated into single cells, only cells with LOH expanded clonally. In contrast, if they were cultured as organoids, cells retained the WT p53 allele. It appears that loss of the WT p53 results in increased resistance to anoikis, which has been shown in a number of studies to increase metastasis of tumours and promote secondary tumors. Thus, the key findings are not surprising and generally follow on from previous work. Importantly, all the effects required the mutant p53, as loss of both p53 alleles did not support metastasis or anoikis resistance.

[...] Interestingly they found that LOH was associated with both increased inflammatory pathway genes, and increased expression of growth factor/MAPK genes. These would certainly fit with the observations in mice, but they haven't really made a functional link in this study.

Response: We thank Reviewer 3 for constructive comments on our manuscript. We have responded to the comments by Reviewer 3 as follows.

Overall, the data are convincing and technically sound as far as I can see. The question is how the gene expression data explains the phenotype, and it is not clear if this link is made. The key gene expressions signatures would fit the finding's. For example, many of the MAPK signalling changes would promote increased resistance to apoptosis. However, they have not directly made this link,

Response: As suggested by Reviewer 3 and Editor, we further examined target molecules of activated upstream regulators in AKTP^{M/LOH} cells by STRING search, and performed pharmacological experiment using MEK inhibitor as follows.

STRING database search indicated significant interaction among target molecules of activated pathways in AKTP^{M/LOH} cells, suggesting that growth factor/MAPK and inflammatory pathways interact to generate transcriptional network in AKTP^{M/LOH} cells. We have added these results in Supplementary Fig. 7 and in the text (page 13, lines 8-11).

Moreover, we examined whether or not MAPK pathway is involved in the acquisition of survival and clonal expansion abilities by treatment of AKTP^{M/LOH} cells with the MEK inhibitor trametinib. Importantly, trametinib treatment significantly suppressed the cloning efficiency from single cells and clonal expansion from 500 cells in 96-well plates. We have now added the results of trametinib treatment as Fig. 7e and 7f and in the text (page 13, lines 12-16). The Methods (page 21, lines 11-12) and Fig 7 legends were also updated accordingly (page 34).

To examine the mechanistic link between inflammatory pathway and increased survival/cloning efficiency, we treated AKTP^{M/LOH} cells with celecoxib, a COX-2 selective inhibitor and aspirin. However, we failed to find suppressive effect on the cell phenotypes. It is possible that inflammatory pathway contributes to increased survival and cloning efficiency in AKTP^{M/LOH} cells through indirect mechanism *in vivo*. Therefore, in the revised manuscript, we have added only the results of trametinib treatment.

Minor points.

In some places they often describe data as a percentage, for example in Figure 2d they state the nuclear p53 index. However, it is not clear how many cells or tumor organoids were analysed to get this nuclear accumulation ratio.

Response: We apologize for not indicating the sample numbers when data were shown as percentages. We have now presented the data as percentages in Fig. 1f, Fig. 2e, f, Fig. 4c and Fig. 5c and e. We have indicated the exact number of organoids, tumor glands and seeded cells in all of these Figures or Figure legends.

Reviewers' comments:

Reviewer #1 (Remarks to the Author):

All my points raised in the previous round of review have been satisfactorily addressed. I have no further concerns.

Reviewer #2 (Remarks to the Author):

The authors have responded well to reviewer criticisms.

Reviewer #3 (Remarks to the Author):

The authors have now performed a few limited experiments to make some mechanistic link between the gene expression signatures and the clonal expansion from single cells. Although it is quite a limited experiment, it is sufficient at this stage, along with changes made in response to the other reviews, to satisfy my reservations about the original manuscript.

Reviewer #4 (Remarks to the Author):

The authors performed a smart design to study how the loss of TP53 WT allele results in increased metastasis of tumours. Transcriptome analyses on different genotypes (A, AKTP+/+, AKTP+/M, AKTPM/LOH and AKTPNull) revealed that the loss of TP53 WT allele by LOH showed significant association with increased inflammatory and growth factor/MAPK pathways which is not observed in AKTPNull cells. The results of this work are relevant, data is well but bioinformatic information is poor and should be improved.

1. How did you perform the demultiplexing of RBNaseq data? which software did you use? Did you remove adapters in this step?
2. Ensembl85 is not a reference build of mouse genome, the authors should change this in the manuscript for an official reference genome like GRCm38.p4/mm10.
3. The authors used for the alignment the software TopHat2 adding some parameters. Why did you choose argument --segment-length 17 and library-type fr-stranded? The segment-length default is 25 but authors changed it to 17. In the guide, it's recommended to decrease the default 25 only with short reads (usually <45-bp) but this work used reads of 50 bp.
4. The bioinformatic protocol used in RNAseq analysis was proposed by Trapnell in 2012 (Differential gene and transcript expression analysis of RNA-seq experiments with TopHat and Cufflinks) but recent publications used STAR, an ultrafast, more sensitive and precise tool that reduces alignment time and increases mapping efficiency. Besides, STAR has the option of two pass mode capable to perform a first run pass of mapping collecting the junctions and use them as annotated junctions for the 2nd pass mapping, increasing even more the alignment efficiency. Why did you use TopHat2?
5. Did you remove PCR duplicates? If not, why did you skip this step?
6. Authors should include a supplementary file with RPKM, log2 fold change, p-values and p-values adjusted by FDR of at least the top signals.
7. Authors should include in supplementary data RNA-seq quality control metrics representations:
 - a. Duplication rates of the reads determined using a sequence-based and a mapping-based strategy. Log2 reads per kb vs Duplication level (%)
 - b. Distribution of read quality scores.
 - c. PCA to visualize if the samples cluster per their conditions.
 - d. Scatter plots like Volcano plots to represent p-value versus fold change

Our point-by-point responses to the comments made by the Reviewers are included below. The comments are quoted **verbatim**, followed by our **responses**. Changes in the manuscript are **highlighted** in the revised text.

Reviewer #1 (Remarks to the Author):

All my points raised in the previous round of review have been satisfactorily addressed. I have no further concerns.

Reviewer #2 (Remarks to the Author):

The authors have responded well to reviewer criticisms.

Reviewer #3 (Remarks to the Author):

The authors have now performed a few limited experiments to make some mechanistic link between the gene expression signatures and the clonal expansion from single cells. Although it is quite a limited experiment, it is sufficient at this stage, along with changes made in response to the other reviews, to satisfy my reservations about the original manuscript.

Response: We thank Reviewers #1, #2 and #3 for expressing their satisfaction with our revised manuscript.

Reviewer #4 (Remarks to the Author):

The authors performed a smart design to study how the loss of TP53 WT allele results in increased metastasis of tumours. Transcriptome analyses on different genotypes (A, AKTP+/, AKTP+/M, AKTPM/LOH and AKTPNull) revealed that the loss of TP53 WT allele by LOH showed significant association with increased inflammatory and growth factor/MAPK pathways which is not observed in AKTPNull cells. The results of this work are relevant, data is well but bioinformatic information is poor and should be improved.

Response: We thank Reviewer #4 for the positive comments on our manuscript. We have responded to all of the comments by Reviewer#4 as follows.

1. How did you perform the demultiplexing of RNAseq data? which software did you use? Did you remove adapters in this step?

Response: During multiplexing, samples are individually labelled with unique indices. The samples are subsequently pooled and sequenced on the same lane. Following sequencing, all reads were demultiplexed on the basis of their unique indices using Illumina's bcl2fastq2 (version 2.20). We have added the method used for demultiplexing of RNA-Seq data in the Methods section (page 23, lines 3-4).

When the single-end of the insert (sonicated fragment size of 200~300 bp) is sequenced with 36/50 cycles, the sequencing reaction starts after the primer binding site at 5'-end. Because the insert length is long, reads of 36-bp / 50-bp cannot reach the adapter at the 3'-end. Thus, we thought that we did not have to remove adapters from the raw data. We have added this information to the Methods section (page 23, line 1-3).

2. Ensembl85 is not a reference build of mouse genome, the authors should change this in the manuscript for an official reference genome like GRCm38.p4/mm10.

Response: We have changed the reference for the alignment of reads from Ensembl85 to GRCm38.p4/mm10, as suggested by Reviewer #4. We have added this information to the Methods section (page 23, lines 12-14).

3. The authors used for the alignment the software TopHat2 adding some parameters. Why did you choose argument --segment-length 17 and library-type fr-stranded? The segment-length default is 25 but authors changed it to 17. In the guide, it's recommended to decreased the default 25 only with short reads (usually <45-bp) but this work used reads of 50 bp.

Response: In this study, we had made full use of the previous gene expression data of A, AKTP^{+/+} and AKTP^{+M} (Sakai et al., Cancer Res, 2018) to detect differentially expressed genes (DEGs) specific to AKTP^{M/LOH} and/or AKTP^{Null} cells. In the study by Sakai et al. (2018), we adjusted the 'segment-length' of TopHat2 ('default 25' to '17') to increase mapping rates and the splice junction rate in sequence alignment with 36-bp of single-end reads that were sequenced in early 2017. In addition, an argument 'read-mismatches' in sequence alignment was changed to '1'. Sequence alignment with segment-length 17 showed slightly higher mapping rate and splice junction rate in comparison to segment-length default 25 (shown in Supplementary Fig. 9); thus 'segment-length 17' was chosen for sequence alignment.

To detect DEGs specific to AKTP^{M/LOH} and/or AKTP^{Null} cells in comparison to A, AKTP^{+/+} and AKTP^{+M}, previous gene expression data of A, AKTP^{+/+} and AKTP^{+M} samples by Sakai et al. (2018) were maintained, and 'segment-length 17' were used for the sequence alignment of AKTP^{M/LOH} and/or AKTP^{Null} samples (50-bp single-end sequence data). Because the mapping rate and splice junction rate were a little low when sequence alignment was performed using TopHat2 with segment-length 17 in comparison to the default, segment-length 25 (shown in Supplementary Fig. 9; showing almost similar distribution between the two values), we thought that the results of this alignment were not a critical issue for the data analysis, and we therefore processed the results of sequence alignment using TopHat2. We have added these explanations in the Methods section (page 23, lines 16-18) and added the data of mapping rate and splice junction rate in Supplementary Fig. 9.

4. The bioinformatic protocol used in RNAseq analysis was proposed by Trapnell in 2012 (Differential gene and transcript expression analysis of RNA-seq experiments with TopHat and Cufflinks) but recent publications used STAR, an ultrafast, more sensitive and precise tool that reduces alignment time and increases mapping efficiency. Besides, STAR has the option of two pass mode capable to perform a first run pass of mapping collecting the junctions and use them as annotated junctions for the 2nd pass mapping, increasing even more the alignment efficiency. Why did you use TopHat2?

Response: As pointed out by the Reviewer #4, we processed the RNA-Seq data on the basis of the Tuxedo protocol reported by Trapnell et al. (Nat Protoc, 2012) using TopHat and Cufflinks. The Tuxedo protocol was used in our previous study (Sakai et al., Cancer Res, 2018). To make full use of previous gene expression data of A, AKTP^{+/+} and AKTP^{+M} as mentioned above, we chose the same method as the previous study of Sakai et al (Cancer Res, 2018) for the RNA-Seq data analysis of AKTP^{M/LOH} and/or AKTP^{Null}. If the sequence alignment using STAR were newly performed, a little more improved, accurate data compared with the results of TopHat2 would be generated. However, we would like to maintain the current presented results by downstream transcriptome analyses and hope to better understand this situation. We are grateful for your informative suggestion related to STAR aligner (Dobin et al., Bioinformatics, 2013), which increases mapping efficiency and improves alignment sensitivity and precision, and we will use the current version for our next project. We have explained the protocol used in this study with citation of new reference by Trapnell et al, (Nat Protoc, 2012) (Ref #41) in the Methods section (page 23, line 11-12).

5. Did you remove PCR duplicates? If not, why did you skip this step?

Response: We skipped the removal of PCR duplicates in data processing because computational removal of read duplicates, especially in low quantity samples can worsen the power for the differential gene expression (Parekh et al. *Sci Rep*, 2016; and Palomares et al. *Sci Rep*, 2019). We have added an explanation about this issue in the Methods section (page 23, lines 7-9) with citation of these additional references (Ref #39 and #40).

The RNA-Seq quality control showed high rates of read duplicates in all samples. The rate of read duplication was determined by sequence- and mapping-based approaches using RSeQC (Wang et al. Bioinformatics, 2012), and the low distribution of uniquely mapped reads from mapping-based QC (11–18%) was observed (Supplementary Fig. 8b). We think that this may result from RNA-Seq library preparation with ultra-low input amounts (especially related to PCR amplification). When we examined the expression patterns of genes related to the p53 signaling pathway, as well as inflammatory pathways, we identified their significant upregulation in AKTP^{+M}, AKTP^{M/LOH} and AKTP^{Null} cells (Supplementary Figure 7). We believe that this reflects the abundance of transcripts in cancer cells even though showing high rates of read duplicates from RNA-Seq data. We have added the QC results in Supplementary Fig. 8b, and described the duplication rate of the reads in the Methods section with citation of a paper by Wang et al. (2012) (page 23, lines 18-22).

6. Authors should include a supplementary file with RPKM, log2 fold change, p-values and p-values adjusted by FDR of at least the top signals.

Response: As suggested by Reviewer #4, we have added Supplementary Table 2, containing (1) the RPKM values of all genes in each replicate of all of analyzed samples and (2) the differential expression (DE) information between different samples including the log2 fold-change, *p*-value and *q*-value. In particular, we have marked DEGs specific to AKTP^{M/LOH} and AKTP^{Null} cells. Information was added to the Methods section (page 24, lines 5-6).

7. Authors should include in supplementary data RNA-seq quality control metrics representations:

a. Duplication rates of the reads determined using a sequence-based and a mapping-based strategy. Log₂ reads per kb vs Duplication level (%)

b. Distribution of read quality scores.

c. PCA to visualize if the samples cluster per their conditions.

d. Scatter plots like Volcano plots to represent p-value versus fold change

Response: According to the comments by the Reviewer #4, we have added RNA-Seq quality control (QC) metrics in Supplementary Figure 8a-d with explanations in the Figure legends. Brief explanations for each quality control metrics were also added to the Methods section.

- Supplementary Fig. 8a shows the distribution of read quality scores with plotting quality scores across all bases at each position in the reads of the samples. The read quality was assessed using FastQC (version 0.11.9) (Andrews et al, Ref #38). (page 23, lines 6-7)

- Supplementary Fig. 8b shows the duplication rates of reads determined by sequence- and mapping-based approaches using RSeQC (Wang et al, 2012, Ref #44). In the plots, the x-axis indicates the duplication time (occurrence) and the y-axis in each approach indicates the number of uniquely mapped reads with log₁₀-transformation (*left*) and the percentage of cumulative (uniquely mapped) reads (*right*), respectively. (page 23, lines 18-22)

- Supplementary Fig. 8c shows PCA results for RNA-Seq samples analyzed. (page 24, lines 2-4)

- Supplementary Fig. 8d shows 'volcano plots' showing fold-change and the *P*-values for the comparisons of AKTP^{+/+} versus AKTP^{M/LOH} (*left*) and AKTP^{+/+} versus AKTP^{Null} (*right*). DEGs (*P*<0.01 and ≥1.5-fold change) are depicted with green-color dots. In particular, DEGs specific to AKTP^{M/LOH} (*left*) and/or AKTP^{Null} (*right*) cells, which were filtered by comparison with other samples, are depicted with red-color dots. Of those cell-specific DEGs, 99.63% and 98.23% were found to be less than *q* < 0.05, respectively (shown in Supplementary Table 2). (page 24, lines 5-7)

REVIEWERS' COMMENTS:

Reviewer #4 (Remarks to the Author):

All my questions have been satisfactorily addressed. The additional information included in the new manuscript help to understand and follow the bioinformatic analysis performed. I have no further concerns.